# Non-canonical autophagy functions of ATG16L1 in epithelial cells limit lethal infection by influenza A virus

Yingxue Wang[1,†], Parul Sharma[2,†], Matthew Jefferson[1,†], Weijiao Zhang[1,†], Ben Bone[1], Anja Kipar[2,3], David Bitto[4], Janine L Coombes[2], Timothy Pearson[1], Angela Man[5] (ID), Alex Zhekova[1], Yongping Bao[1], Ralph A Tripp[6], Simon R Carding[1,7] (ID), Yohei Yamauchi[4], Ulrike Mayer[8], Penny P Powell[1] (ID), James P Stewart[2,6,*] (ID) & Thomas Wileman[1,7,**] (ID)

## Abstract

Influenza A virus (IAV) and SARS-CoV-2 (COVID-19) cause pandemic infections where cytokine storm syndrome and lung inflammation lead to high mortality. Given the high social and economic cost of respiratory viruses, there is an urgent need to understand how the airways defend against virus infection. Here we use mice lacking the WD and linker domains of ATG16L1 to demonstrate that ATG16L1-dependent targeting of LC3 to single-membrane, non-autophagosome compartments – referred to as non-canonical autophagy – protects mice from lethal IAV infection. Mice with systemic loss of non-canonical autophagy are exquisitely sensitive to low-pathogenicity IAV where extensive viral replication throughout the lungs, coupled with cytokine amplification mediated by plasmacytoid dendritic cells, leads to fulminant pneumonia, lung inflammation and high mortality. IAV was controlled within epithelial barriers where non-canonical autophagy reduced IAV fusion with endosomes and activation of interferon signalling. Conditional mouse models and ex vivo analysis showed that protection against IAV infection of lung was independent of phagocytes and other leucocytes. This establishes non-canonical autophagy in airway epithelial cells as a novel innate defence that restricts IAV infection and lethal inflammation at respiratory surfaces.

**Keywords** ATG16L1 WD Domain; cytokine storm; influenza; intrinsic defence; non-canonical autophagy
**Subject Categories** Autophagy & Cell Death; Microbiology, Virology & Host Pathogen Interaction

The EMBO Journal (2021) 40: e105543

## Introduction

Influenza A virus (IAV) is a respiratory pathogen of major global public health concern (Yamayoshi & Kawaoka, 2019). As with SARS-CoV-2, animal reservoirs of IAV can contribute to zoonotic infection leading to pandemics with a high incidence of viral pneumonia, morbidity and mortality. IAV infects airway and alveolar epithelium and damage results from a combination of the intrinsic pathogenicity of individual virus strains as well as the strength and timing of the host innate/inflammatory responses. Optimal cytokine levels protect from IAV replication and disease but excessive cytokine production and inflammation worsens the severity of lung injury (Davidson *et al*, 2014; Iwasaki & Pillai, 2014; Teijaro *et al*, 2014; Herold *et al*, 2015; Ramos & Fernandez-Sesma, 2015). Even though infection of the lower respiratory tract can result in inflammation, flooding of alveolar spaces, acute respiratory distress syndrome and respiratory failure, the factors that control IAV replication at epithelial surfaces and limit lethal lung inflammation remain largely unknown.

The transport of viruses to lysosomes for degradation provides an important barrier against infection. Transport to lysosomes can be enhanced by non-canonical autophagy pathways which conjugate autophagy marker protein LC3 to endo-lysosome compartments to increase lysosome fusion. In phagocytes, LC3-associated phagocytosis (LAP) conjugates LC3 to phagosomes and enhances phagosome maturation (Sanjuan *et al*, 2007; Delgado *et al*, 2008; Martinez *et al*, 2015; Lamprinaki *et al*, 2017; Fletcher *et al*, 2018). In non-

1   Norwich Medical School, University of East Anglia, Norwich, UK
2   Department of Infection Biology and Microbiomes, University of Liverpool, Liverpool, UK
3   Institute of Veterinary Pathology, University of Zurich, Zurich, Switzerland
4   School of Cellular and Molecular Medicine, Faculty of Life Sciences, University of Bristol, Bristol, UK
5   Earlham Institute, Norwich, UK
6   Department of Infectious Disease, University of Georgia, Georgia, USA
7   Gut Microbes and Health Research Programme, Quadram Institute Bioscience, Norwich, UK
8   School of Biological Sciences, University of East Anglia, Norwich, UK
    *Corresponding author. Tel: +44 151 795 0221; E-mail: j.p.stewart@liv.ac.uk
    **Corresponding author. Tel: +44 1603 591238; E-mail: t.wileman@uea.ac.uk
    †These authors contributed equally to this work.

phagocytic cells, LC3 is conjugated to endo-lysosome compartments during the uptake of particulate material such as apoptotic cells and aggregated β-amyloid, and following membrane damage during pathogen entry or osmotic imbalance induced by lysosomotropic drugs (Florey et al, 2011, 2015; Roberts et al, 2013; Tan et al, 2018; Heckmann et al, 2019). It is known from in vitro studies that LC3 can be recruited to endo-lysosome compartments during the uptake of pathogens, but the roles played by non-canonical autophagy during viral infection in vivo are largely unknown.

A role for non-canonical autophagy in host defence has been implied from in vitro studies of LAP in phagocytes infected with free living microbes with a tropism for macrophages such as bacteria (Listeria monocytogenes (Gluschko et al, 2018), Legionella dumoffii (Hubber et al, 2017)), protozoa (Leishmania major) and fungi (Aspergillus fumigatus (Akoumianaki et al, 2016, Kyrmizi et al, 2018, Matte et al, 2016)). It is also known that IAV induces non-canonical autophagy during infection of cells in culture (Fletcher et al, 2018); however, the role played by non-canonical autophagy in controlling IAV infection and lung inflammation in vivo is currently unknown. It is not known, for example, whether non-canonical autophagy is important in the control of IAV infection by epithelial cells at sites of infection or whether it plays a predominant role within phagocytes and antigen-presenting cells during development of an immune response. Herein we use mice with specific loss of non-canonical autophagy to determine the role played by non-canonical autophagy in host defence against IAV infection of the respiratory tract. The mice (δWD) lack the WD and linker domains of ATG16L1 that are required for conjugation of LC3 to endo-lysosome membranes (Rai et al, 2019) but express the N-terminal ATG5-binding domain and the coiled coil domain (CCD) and linker residues up to glutamate at position 230 (E230) of ATG16L1 that are required for WIPI2 binding and autophagy (Dooley et al, 2014). Importantly, the δWD mice grow normally and maintain tissue homeostasis (Rai et al, 2019), and unlike mice with LysMcre-mediated deletion of autophagy genes from myeloid cells (Lu et al, 2016), or disruption of ATG16L1 through loss of the CCD (Saitoh et al, 2008), the δWD mice do not have a pro-inflammatory phenotype.

We show that loss of non-canonical autophagy from all tissues renders mice highly sensitive to low-pathogenicity murine-adapted IAV (A/X-31) leading to extensive viral replication throughout the lungs, cytokine dysregulation and high mortality typically seen after infection with highly pathogenic IAV. Conditional mouse models and ex vivo analysis showed that protection against IAV infection of lung was independent of phagocytes and other leucocytes, and that infection was controlled within epithelial barriers where non-canonical autophagy slowed fusion of IAV with endosomes and reduced interferon signalling. This establishes non-canonical autophagy pathways in airway epithelial cells as a novel innate defence mechanism that restricts IAV infection at respiratory surfaces.

## Results

### Mice with systemic loss of the WD and linker domains of ATG16L1 are highly sensitive to IAV infection

Panels A and B of Fig EV1 show the rationale for removing the WD and linker domains from ATG16L1 to generate mice (δWD) with a specific loss of non-canonical autophagy (E230 mice described in

Rai et al, 2019). The consequences of loss of the WD and linker domains of ATG16L1 on conventional autophagy and non-canonical autophagy were confirmed using cell lines taken from controls and δWD mice. Figure 1A shows that mouse embryo fibroblasts (MEFs) from littermate control mice recruited LC3 to small puncta indicative of autophagosomes when they were starved in HBSS to induce conventional autophagy and recruited LC3 to large endo-lysosomal vacuoles when non-canonical autophagy was induced by chloroquine or monensin. HBSS induced LC3 puncta in MEFs from δWD mice, but the MEFs were unable to recruit LC3 to large vacuoles induced by chloroquine or monensin. LC3 recruitment was quantified by imaging LC3-positive puncta and vacuoles. The graphs (Fig 1B) show that numbers of autophagosomes induced by HBSS were the same in each cell type but MEFs from δWD were unable to recruit LC3 to large vacuoles when incubated with chloroquine or monensin. The results indicated a selective loss of non-canonical autophagy. LC3 recruitment was quantified by Western blot to detect LC3II, the lipidated form of LC3 that binds membranes. Figure 1C shows that control MEFs expressed the full-length α and β forms of ATG16L1 at 70 kDa and generated increased levels of LC3II following starvation in HBSS or incubation with monensin or chloroquine. MEFs from δWD mice expressed a truncated ATG16L1 at 30 kDa (Fig 1C). The LC3II signal in δWD cells increased after starvation to levels that were similar to starvation controls. LC3II also increased in δWD MEFs incubated with monensin and chloroquine. Monensin and chloroquine raise lysosomal pH, and the consequent inhibition of proteolysis slows the efflux of amino acids from the lysosome. This in turn inhibits the Ragulator-Rag:MTORC1 complex and induces autophagy. Previous work (Fletcher et al, 2018) has shown that monensin activates conventional autophagy and at the same time raised lysosomal pH slows fusion of autophagosomes with lysosomes. This explains the accumulation of small LC3 puncta and increased LC3II observed in δWD cells incubated with monensin or chloroquine. Quantification of Western blots (Fig 1D) showed that there was no significant difference between LC3II signals in control and δWD MEFs after starvation suggesting that autophagy was equally active in the two cell types. The LC3II signal in δWD MEFs incubated with monensin or chloroquine was however lower than controls but not significantly different. Studies in phagocytic cells have shown that non-canonical autophagy/LAP is downstream of Rubicon and PHOX:NOX2 ROS signalling (Martinez et al, 2015). Addition of diphenyliodonium (DPI), an inhibitor of NOX2, to cells incubated with monensin or chloroquine inhibited recruitment of LC3 to large vacuoles (Figure EV1C) indicating that WD domain-dependent non-canonical autophagy is also downstream of ROS signalling in non-phagocytic cells. Microscopy and line profile analysis were used to determine whether recruitment of LC3 to phagosomes in bone marrow-derived macrophages (BMDM) induced by Zymosan also required the WD domain of ATG16L1. Figure 1E-H shows that LC3 was recruited to phagosomes in control BMDM, but was not recruited to phagosomes in BMDM from δWD mice. Taken together, the results show that the δWD mice have specific loss of non-canonical autophagy in both myeloid and non-myeloid cells.

IAV enters airway and lung epithelial cells by endocytosis, and in tissue culture IAV induces non-canonical autophagy leading to ATG16L1-WD domain-dependent conjugation of LC3 to the plasma membrane and peri-nuclear structures (Fletcher et al, 2018). To test

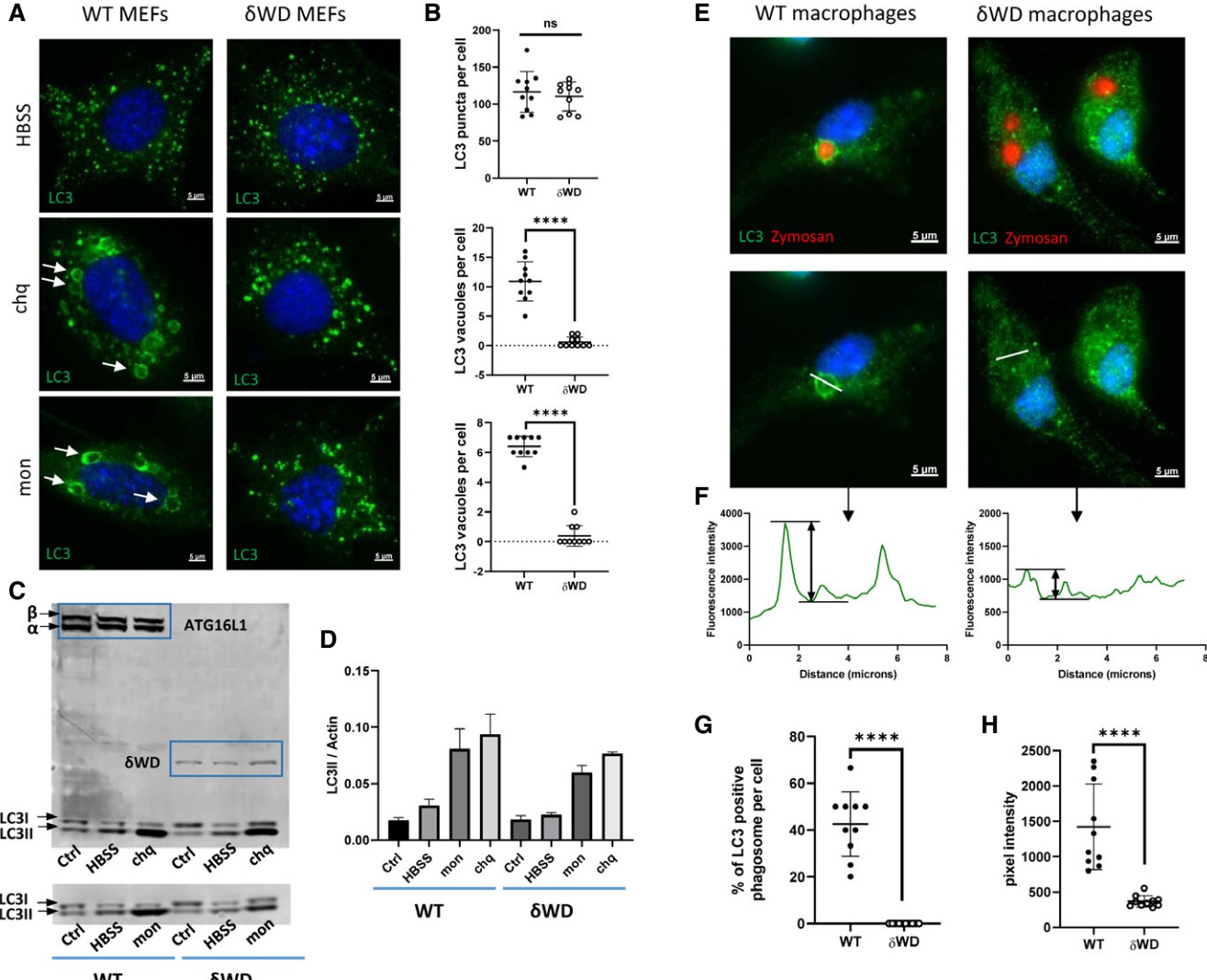

**Figure 1. Deletion of the WD and linker domains of ATG16L1 leads to selective loss of non-canonical autophagy.**

A   MEFs from δWD and littermate control mice were incubated with HBSS, chloroquine or monensin for 2 h as indicated. Cells were fixed and permeabilised and stained for LC3. Arrows indicate LC3 vacuoles where a ring of LC3 signal surrounds vacuoles ranging between 2 and 8 μm diameter.

B   Numbers of fluorescent LC3 puncta and LC3 vacuoles in each cell were quantified by fluorescence microscopy. LC3 puncta were identified using spot function software to locate puncta ranging from 0.5 to 1.0 μm diameter. LC3 vacuoles were identified by eye as rings of fluorescence ranging between 2 and 8 μm diameter. Data from 10 cells are shown, and bars represent the mean ± SD and were compared by Student's *t*-test (*$P < 0.05$, ****$P < 0.0001$).

C–E   MEFs from δWD and littermate control mice were incubated with HBSS, chloroquine or monensin for 2 h as indicated and cell lysates analysed by Western blot for ATG16L1, δWD and LC3 as indicated. Control MEFs express α and β isoforms of ATG16L1 at 70 kDa, and MEFs from δWD mice express a truncated ATG16L1 at 30kDa. (D) shows the level of conversion of LC3 to LC3II estimated by densitometry from a mean (±SD) of three replicate blots. (E) shows fluorescence images of phagosomes following engulfment of Zymosan (red) by bone marrow-derived macrophages (BMDM) from control and δWD mice. White line indicates track used for line profile analysis to compare the LC3 signal on the limiting membrane of the phagosome with the centre.

F   Shows examples of line profile analysis

G, H   (G) shows the percentage of LC3-positive phagosomes per cell, and (H) shows line profile analysis of data from 10 cells, and bars represent the mean ± SD and were compared by Student's *t*-test (****$P < 0.0001$).

whether non-canonical autophagy has a host defence function *in vivo*, δWD mice were infected with IAV. We used a low-pathogenicity murine-adapted IAV (A/X31) that does not normally lead to extensive viral replication throughout the lungs, or cause the cytokine storm syndrome and death typically seen after infection with highly pathogenic viral strains. The results (Fig 2) showed that

δWD mice became moribund and showed severe signs of clinical illness (rapid breathing, piloerection). They also displayed rapid weight loss compared to littermate controls (Fig 2A) and had increased mortality with survivors recovering more slowly from infection (Fig 2B). Virus titres in the lungs of both mice increased with time (Fig 2C) and increased weight loss in δWD mice was

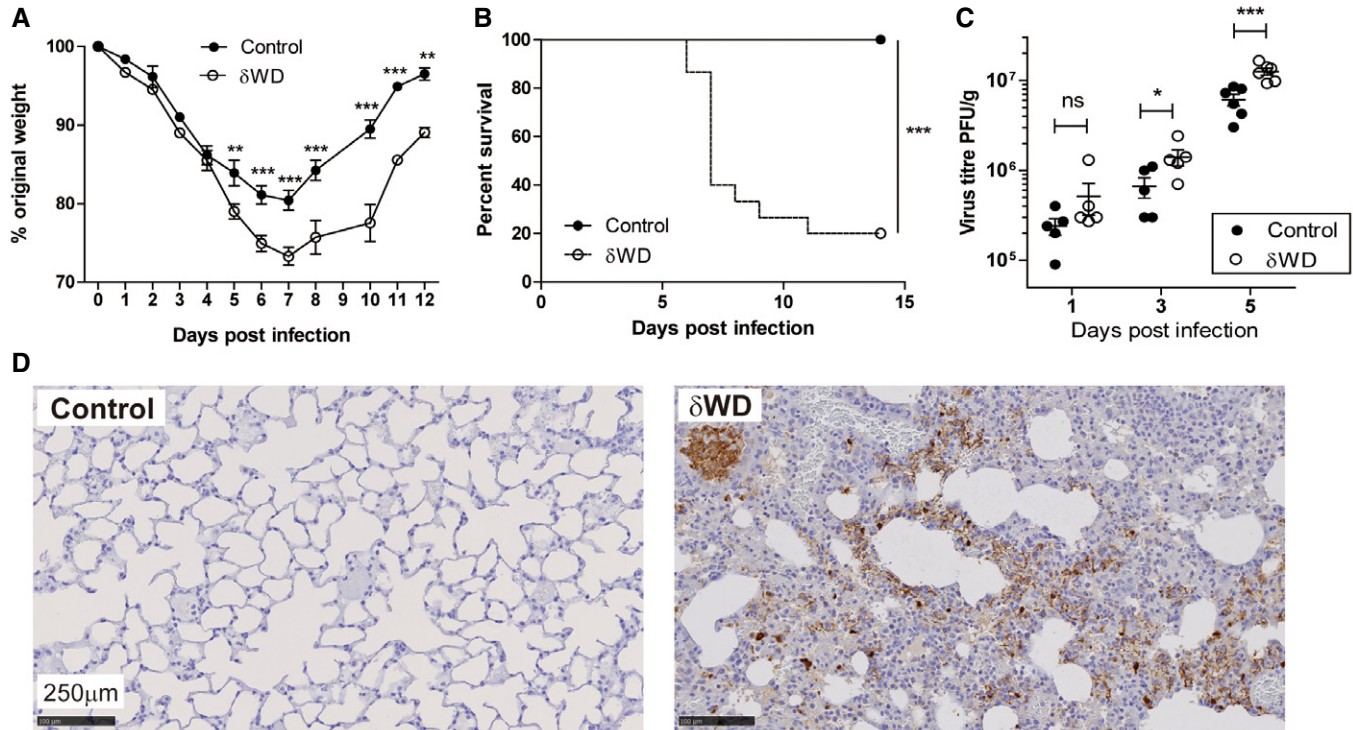

**Figure 2.  Systemic loss of non-canonical autophagy increases susceptibility to IAV infection.**

A–D   Littermate control and δWD mice were challenged intranasally with IAV strain X31 ($10^3$ pfu). (A) Mice were monitored for weight loss at indicated time points.
(n = 8). Data represent the mean value ± SEM. Comparisons were made using a repeated-measures two-way ANOVA (Bonferroni post-test (**$P < 0.01$,
***$P < 0.001$)). (B) Survival was assessed at indicated time points (n = 15). Comparisons were made using log-rank (Mantel–Cox) test ***$P < 0.001$. (C) IAV titre in
lungs was determined by plaque assay at 5 d.p.i. (n = 6). Data for individual animals are shown, and bars represent the mean ± SD. Mann–Whitney $U$-test was
used to determine significance (**$P < 0.01$. (D) The presence of IAV antigen was assessed by IH at 7 d.p.i. (representative images from n = 6).

associated with an approx. log increase in lung virus titre at 5 days post-infection (d.p.i). Furthermore, histopathology and immunohistochemistry (IH) analysis of lungs from δWD mice showed fulminant viral pneumonia with large numbers of IAV-positive cells (Fig 2D). Lungs from control and δWD mice did not show signs of inflammation before infection (Fig EV2).

**Non-canonical autophagy controls lung inflammation after IAV infection**

Innate protection against IAV is provided by type 1 (α, β) and III (λ) interferon (IFN) with severe IAV infection causing excessive airway inflammation and pulmonary changes attributable in part to IFNαβ and TNF-α (Szretter *et al*, 2007; Davidson *et al*, 2014). Measurement of cytokine expression at 2 d.p.i showed that IAV induced a transient increase in transcripts for interferon-stimulated genes (ISGs), ISG15 and IFIT1 (Iwasaki & Pillai, 2014) and pro-inflammatory cytokines (IL-1β, TNF-α and CCL2 [MCP-1]) in the lungs of both control and δWD mice (Fig 3A). This increase in cytokine expression was resolved by 3 d.p.i. before a second wave of increased cytokine expression at 5 d.p.i. This second wave of cytokine expression was resolved by 7 d.p.i in control mice, but δWD mice showed sustained increases in ISG15, IFIT1, IL-1β, TNF-α and CCL2 transcripts, coincident with exacerbated weight loss. At 3 d.p.i, lungs of

δWD mice showed increased expression of neutrophil chemotaxis factor CXCL1 mRNA (Fig 3A), coincident with increased neutrophil infiltration of airways and parenchyma, and extensive neutrophil extracellular traps (NETs) as a consequence of neutrophil degeneration as shown by IH (Fig 3B and Appendix Fig S1). Increased neutrophil infiltration of airways in δWD mice at 2 d.p.i. was confirmed and quantified using flow cytometric analysis of bronchoalveolar lavage (BAL; Fig 3C). At 5–7 d.p.i., increased expression of CCL2 mRNA in δWD mice was coincident with extensive macrophage/monocyte infiltration into lung parenchyma observed by IH (Fig 3B and Appendix Fig S2) which was not seen in controls or the lungs of δWD mice before infection with virus (EV2). This increased macrophage/monocyte infiltration in δWD mice was confirmed and quantified using flow cytometric analysis of single-cell suspensions from lung tissue (Fig 3D). It is known that, in severe IAV infection, a cytokine storm occurs that is amplified by plasmacytoid dendritic cells (pDCs) (Davidson *et al*, 2014). pDCs detect virus-infected cells and produce large amounts of cytokines, in particular IFNαβ, that in severe infections can enhance disease. In these cases, depletion of pDCs can decrease morbidity (Davidson *et al*, 2014). Depletion of pDCs in IAV-infected δWD mice using anti-PDCA-1 led to markedly decreased weight loss as compared with isotype control-treated mice and that was similar to that seen in littermate controls (Fig 3E). This indicates that excessive cytokine

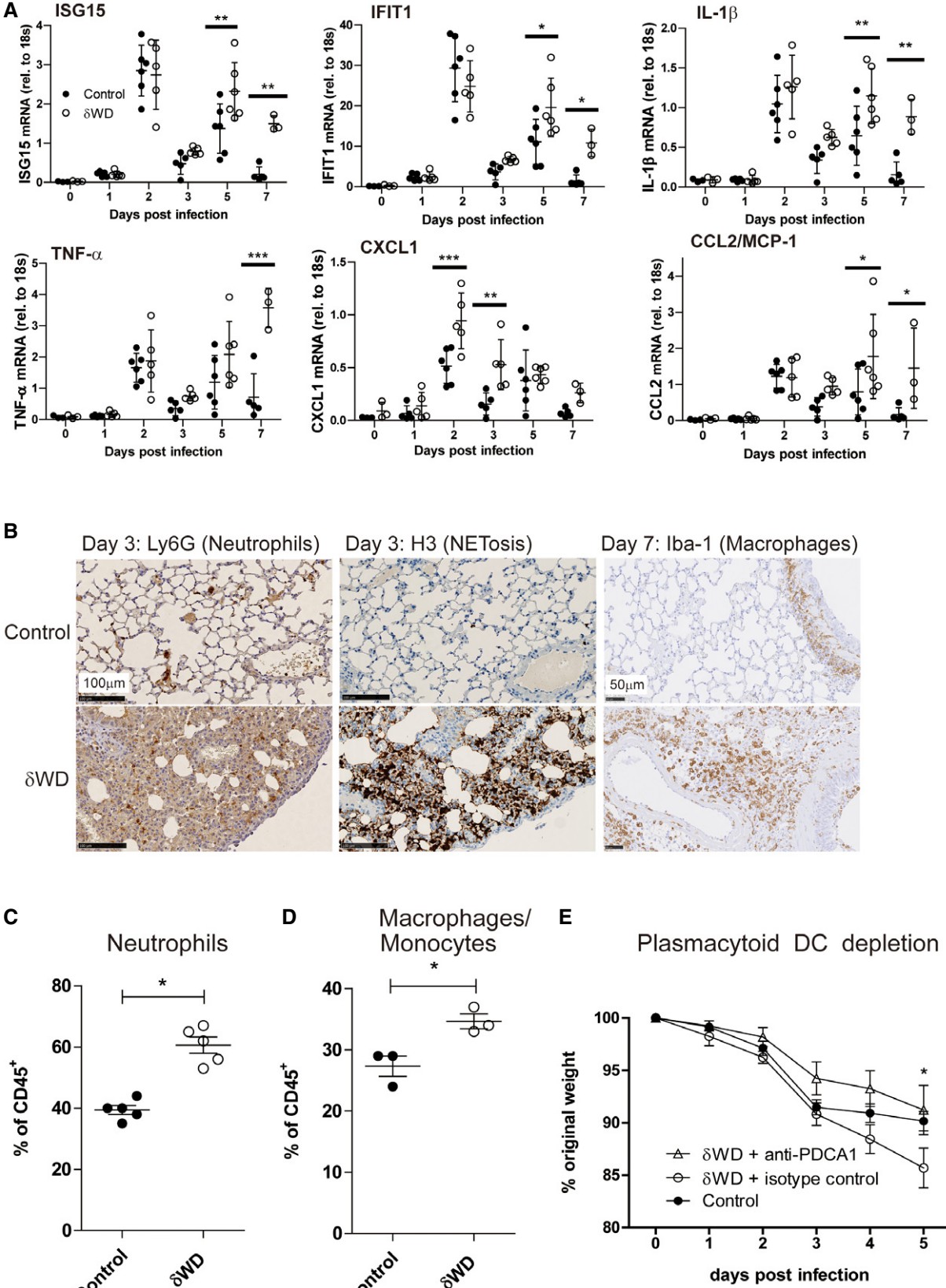

**Figure 3.**

**Figure 3. Systemic loss of non-canonical autophagy leads to extensive lung inflammation and damage.**

A–E Littermate control and δWD mice (*n* = 5) were challenged with IAV X31 ($10^3$ pfu). (A) At the indicated time points, cytokine mRNA transcripts in lung tissue (*n* = 5) were evaluated by qPCR. Data for individual animals are shown, and bars represent the mean ± SD and were compared by 2-way ANOVA with Bonferroni post-tests (*$P$ < 0.05, **$P$ < 0.01, ***$P$ < 0.001). (B) Representative lung sections from animals (*n* = 6) taken at 3 d.p.i. were stained by IH for neutrophils (Ly6G) or neutrophil extracellular traps (NET; anti-H3). Sections at 7 d.p.i. were stained for macrophages (Iba-1). Further micrographs are shown in Appendix Figs S1 and S2. (C) BAL (*n* = 5) was taken at 2 d.p.i and evaluated by flow cytometry, with pre-gating on CD45⁺. The percentage of neutrophils (CD11b⁺, Ly6G⁺) cells is shown ± SEM and was compared using Mann–Whitney *U*-test (*$P$ < 0.05). (D) Single-cell suspensions were prepared from lungs taken at 5 d.p.i and evaluated by flow cytometry, with pre-gating on CD45⁺. The percentage of macrophage/monocytes (CD11b⁺, F4/80⁺) cells is shown ± SEM and was compared using Mann–Whitney *U*-test (*$P$ < 0.05). (E) δWD mice were treated with either anti-PCDA-1 (to deplete plasmacytoid DC) or an isotype-matched control. Littermate control mice were used as comparator. Weight loss was measured at the indicated days p.i. (*n* = 5). Comparisons were made using a repeated-measures two-way ANOVA (Bonferroni post-test, *$P$ < 0.05).

production amplified by pDCs is responsible for the increased morbidity seen in the δWD mice.

Thus, mice with systemic loss of non-canonical autophagy failed to control lung virus replication and inflammation, leading to increased cytokine production, morbidity and mortality.

## Systemic loss of the WD and linker domains of ATG16L1 does not lead to gross changes in inflammatory threshold or immunological homeostasis

Macrophages cultured from embryonic livers from mice with loss of the coiled coil domain of ATG16L1 are unable to activate canonical autophagy and secrete high levels of IL-1β (Saitoh *et al*, 2008). Similarly, LysMcre-mediated deletion of genes essential for conventional autophagy (e.g. Atg5, Atg7, Atg14, Atg16L1, FIP200) in mice leads to raised pro-inflammatory cytokine expression in the lung. This has been reported to increase resistance to IAV infection (Lu *et al*, 2016), and this was also observed in mice used in our study (Appendix Fig S3) where LysMcre-mediated loss of Atg16L1 resulted in weight loss similar to controls (Appendix Fig S3A) and reduced overall virus titre (Appendix Fig S3B). This led us to test the possibility that the δWD mutation to ATG16L1 could also increase IL-1β secretion and cause the increased inflammation observed during IAV infection. This was tested by incubating BMDM with LPS and purine receptor agonist, BzATP (Appendix Fig S4A), or by challenging mice with LPS (Appendix Fig S4B). Mice with a complete loss of ATG16L1 in myeloid cells (Atg16L1^fl/fl-lysMcre) showed threefold increases in IL-1β in serum and increased secretion IL-1β from BMDM *in vitro*. In contrast IL-1β secretion in δWD mice did not differ significantly from littermate controls (Appendix Fig S4A and B). This was consistent with lack of elevated cytokines in lungs prior to infection (see day 0 in Fig 3A), and our previous work shows that serum levels of IL-1β, IL-12p70, IL-13 and TNF-α in δWD mice are the same as in littermate controls at 8-12 and 20-24 weeks (Rai *et al*, 2019). The exaggerated inflammatory response to IAV in δWD mice did not therefore result from a raised pro-inflammatory threshold or dysregulated IL-1β responses in the lung. Also, the frequencies of T cells, B cells and macrophages were similar in δWD mice to littermate controls (Appendix Fig S5). These data suggest that the exaggerated responses of δWD mice to IAV do not occur because the mice have a raised inflammatory threshold or abnormal immunological homeostasis.

## Non-canonical autophagy limits IAV infection independently of phagocytic cells

The link between non-canonical autophagy, TLR signalling, NADPH oxidase activation and ROS production (Sanjuan *et al*, 2007;

Delgado *et al*, 2008; Martinez *et al*, 2015) provides phagocytes with a powerful mechanism to limit infections *in vivo*. To test whether wild-type bone marrow-derived cells could protect susceptible δWD mice from lethal IAV infection, we generated radiation chimaeras (Fig EV3). When challenged with IAV, δWD mice reconstituted with either wild-type or δWD bone marrow remained highly sensitive to IAV (Fig 4A and B) with body weight reduced by up to 25% and decreased survival by 5 d.p.i. As seen for δWD mice, weight loss was associated with a 10-fold increase in lung viral titre (Fig 4C), fulminant pneumonia and inflammatory infiltration into the lung (Fig 4D). This increased susceptibility to IAV was not observed for control mice reconstituted with wild-type bone marrow, showing that non-canonical autophagy pathways in phagocytes and other leucocytes from control mice were not able to protect δWD mice against lethal IAV infection. In a reciprocal experiment (Fig 5), mice expressing Cre recombinase in myeloid cells (LysMcre) were used to generate mice (called δWD^phag), where the truncated *Atg16*L1δWD gene was restricted to phagocytic cells (Appendix Fig S6). In these mice, non-canonical autophagy was absent in cultured phagocytes (BMDM) but it was present in skin fibroblasts (Appendix Fig S6E). After infection with IAV, δWD^phag mice showed comparable weight loss and virus titres to those seen in littermate control mice (Fig 5A and B). Likewise, the raised IL-1β levels (Fig 5C) and profuse macrophage and neutrophil lung infiltration observed in δWD mice were absent (Fig EV4) and similar to littermate controls. The ability of the WD and linker domains of ATG16L1 to protect epithelial cells against IAV infection was tested *ex vivo* to further exclude any contribution from recruited leucocytes. Virus titres in precision-cut lung slices (Fig 5D) increased over 3 days and similar to the kinetics seen in vivo titres from δWD mice rose to 10-fold greater than controls. Thus, the sensitivity of δWD mice to IAV was not due to the loss of non-canonical autophagy from myeloid cells, making it likely that non-canonical autophagy mediated by the WD and linker domains of ATG16L1 protects against lethal IAV infection in non-myeloid tissue.

## Non-canonical autophagy slows IAV fusion with endosomes and reduces interferon signalling

IAV enters cells by receptor-mediated endocytosis where acidification of late endosomes results in fusion with the endosomal membrane and delivery of viral ribonuclear proteins (RNPs) into the cytoplasm (Wharton *et al*, 1994; Skehel & Wiley, 2000). RNPs are then imported into the nucleus for genome replication (Boulo *et al*, 2007). Figure 6A shows that the generation of infectious virus was greater in δWD MEFs compared to control. IAV binding was analysed using fluorescent virus, and Fig 6B shows that binding

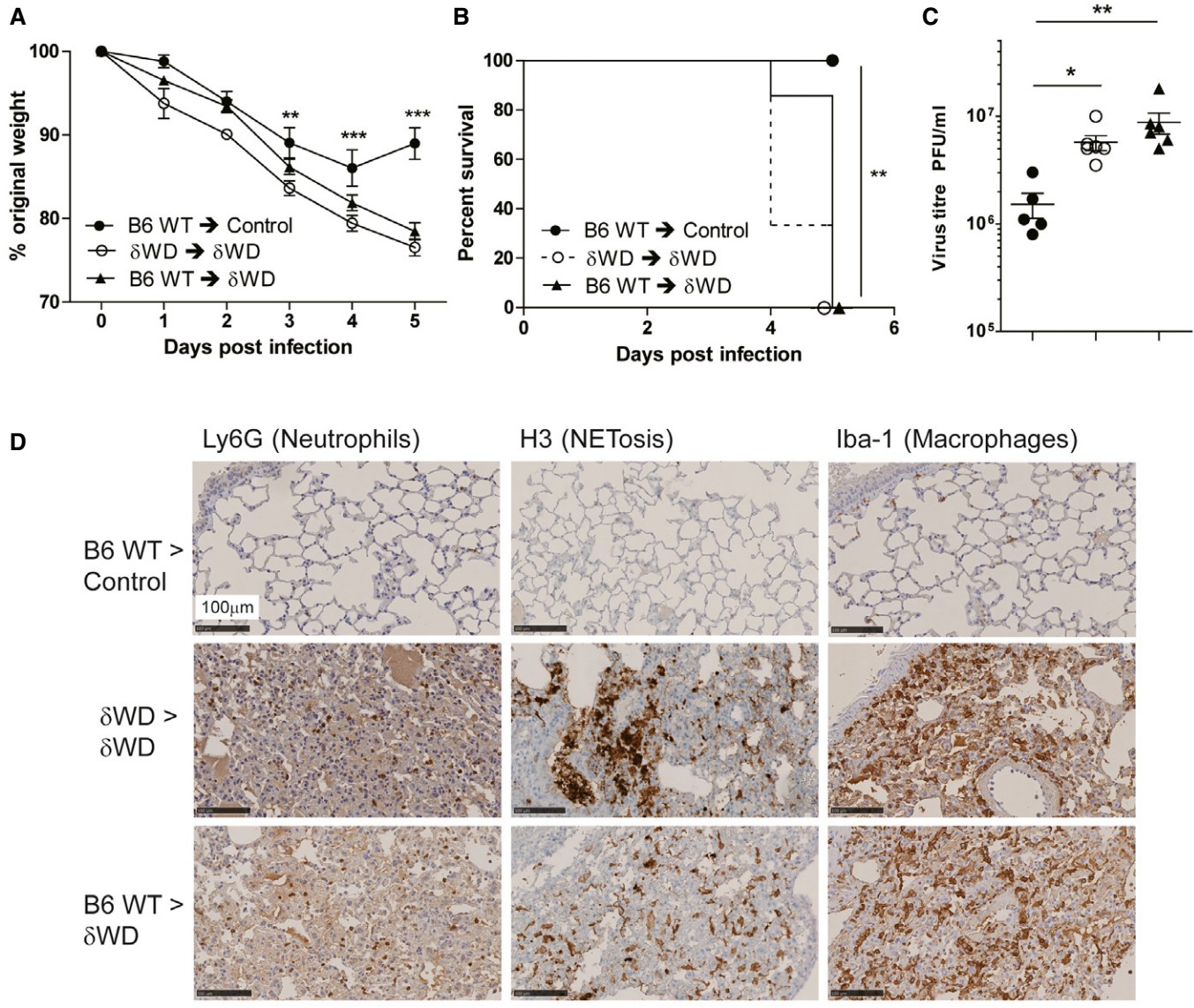

**Figure 4. Bone marrow-derived lymphoid tissue cannot reverse sensitivity to IAV infection.**

A–D   Bone marrow from wild-type (*Atg16*L1$^{+/+}$) was used to reconstitute irradiated littermate control mice (B6 WT → control [●]) or δWD mice (B6 WT → δWD [O]).
Bone marrow from δWD mice was used to reconstitute irradiated δWD mice (δWD → δWD [▲]). After 12 weeks, mice (*n* = 5 per group) were challenged with IAV
X31 (10³ pfu). (A) Mice were monitored for weight loss at indicated time points. Data represent the mean value ± SEM. Comparisons were made using a repeated-
measures two-way ANOVA (Bonferroni post-test, **$P$ < 0.01, ***$P$ < 0.001). (B) Survival was assessed at indicated time points. Comparisons were made using log-
rank (Mantel–Cox) test **$P$ < 0.01. (C) IAV titre in lungs was determined by plaque assay at 5 d.p.i. (*n* = 6). Data for individual animals are shown using symbols
described in (A) and (B), and bars represent the mean ± SD. A one-way ANOVA with Tukey's post hoc analysis was used to determine significance (*$P$ < 0.05,
**$P$ < 0.01). (D) Lungs taken at 5 d.p.i. were analysed for neutrophils (Ly6G), neutrophil extracellular traps (NET; anti-H3) and macrophages (Iba-1).

was similar between control and δWD MEFs suggesting that subsequent steps of endocytosis and viral fusion may be increased in δWD MEFs. The possibility that the WD domain of ATG16L1 affected replication of IAV independently of endocytosis was tested using an acid bypass assay. IAV was bound to cells at 4° and warmed to 37°C for 2 min at pH 6.8 (control) or pH 5 to induce direct fusion with the plasma membrane. When early stages of replication were assessed by staining for nuclear protein, there was no difference between δWD MEFs and control (Fig 6C). This made it

unlikely that the WD domain has a direct role in facilitating IAV replication. The effect of non-canonical autophagy on IAV entry was tested using fluorescence de-quenching assay where the envelope of purified IAV was labelled with green (DiOC18) and red (R18) lipophilic dyes. Individual fusion events in cells estimated by automated confocal microscopy (Fig 6D) show that the number of fusion events per cell was increased after 60 min at 37°C in δWD MEFs compared to control. Similarly, FACS analysis (see also Fig EV5) of the percentage of cells with de-quenched signal showed greater

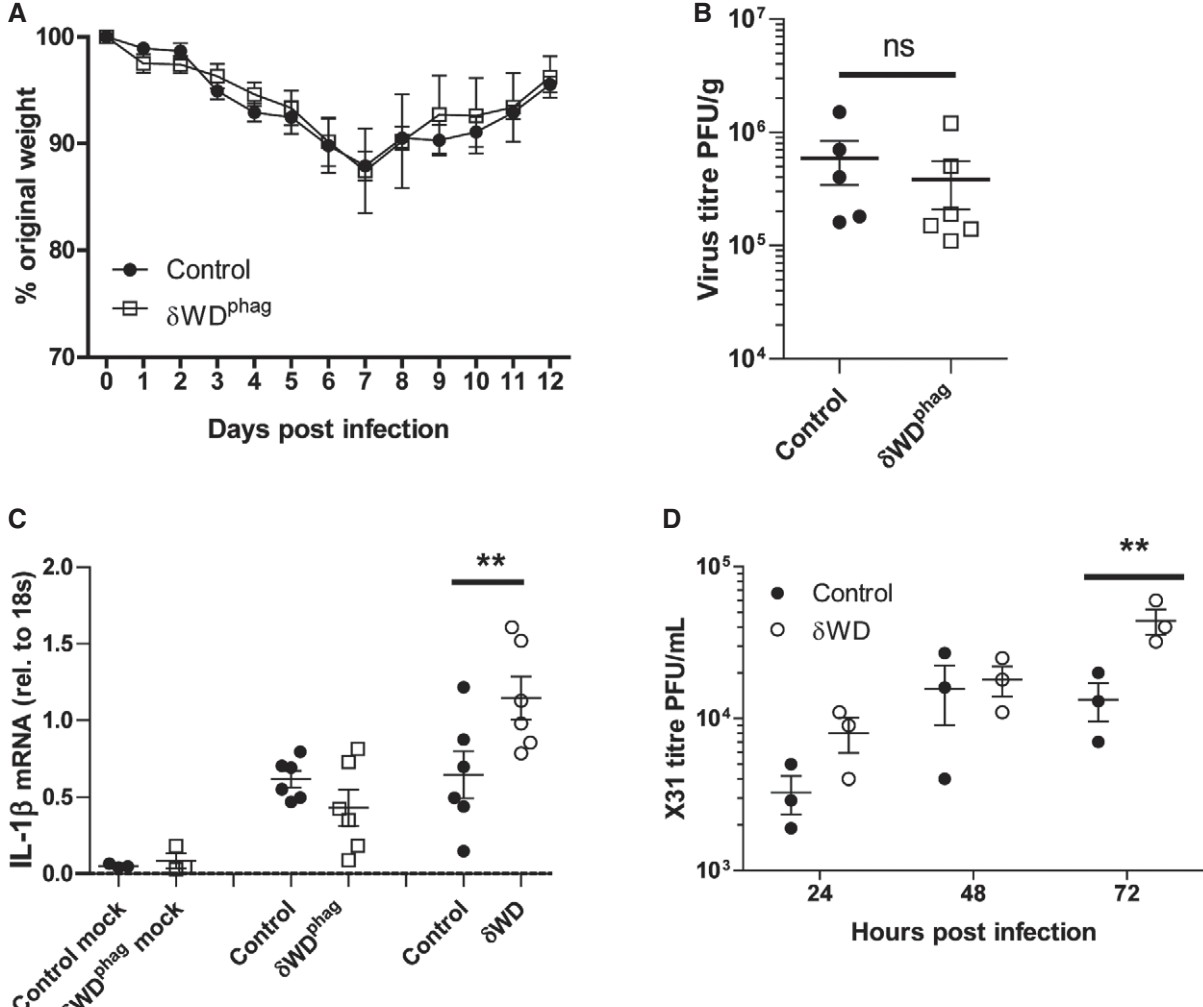

**Figure 5. Loss of non-canonical autophagy from phagocytes does not increase sensitivity to IAV infection.**

A–C δWD^phag mice lack non-canonical autophagy in myeloid (LysMcre) cells (for construction see Appendix Fig S6A). Offspring negative for LysMcre were used as littermate controls. Mice (n = 6 per group) were challenged intranasally with IAV X31 (10³ pfu). (A) Mice were monitored for weight loss at indicated time points. Data represent the mean value ± SEM. Comparisons were made using a repeated-measures two-way ANOVA (Bonferroni post-test). (B) IAV titre in lungs was determined by plaque assay at 5 d.p.i. (n = 6). Data for individual animals are shown, and bars represent the mean ± SD. Mann–Whitney U-test was used to determine significance. (C) IL-1β mRNA transcripts in lung at 5 d.p.i. were determined by qPCR. Mann–Whitney U-test was used to determine significance (**P < 0.01).

D Precision-cut lung slices from control and δWD mice were infected with IAV. Virus titres were determined at indicated time points. Comparisons were made using two-way ANOVA with Bonferroni post-tests (**P < 0.01).

fusion in δWD MEFs (60%) compared to controls (40%) at 30 min and this increased at 60 min (73% versus 56% for controls; Fig 6E), as did mean fluorescence intensity (Fig 6F). Endocytosed viruses were also estimated by automated confocal microscopy (Fig 6 G and H) of permeabilised cells and again there was a significant increase in endocytosis of IAV in δWD MEFs at 30 min. Taken together, the results showed that the WD domain of ATG16L1 slowed fusion of IAV with endosome membranes. The recognition of viral RNA by interferon sensors following delivery of RNPs into the cytoplasm was used as a second assay for IAV entry. MEFs from δWD mice showed between three- and fivefold increases in expression of IFN responsive genes, ISG15 and IFIT1 (Fig 6I and J), and this was also

observed in the lung *in vivo* (Fig 3A). Taken together, the results demonstrate for the first time that the WD and linker domains of ATG16L1 allow non-canonical autophagy to provide a novel innate defence mechanism against lethal IAV infection within the epithelial barrier *in vivo*.

# Discussion

Respiratory viruses such as influenza A virus (IAV) and SARS-CoV-2 can move from animal reservoirs to create human pandemics with high morbidity and mortality. The danger posed by pandemic

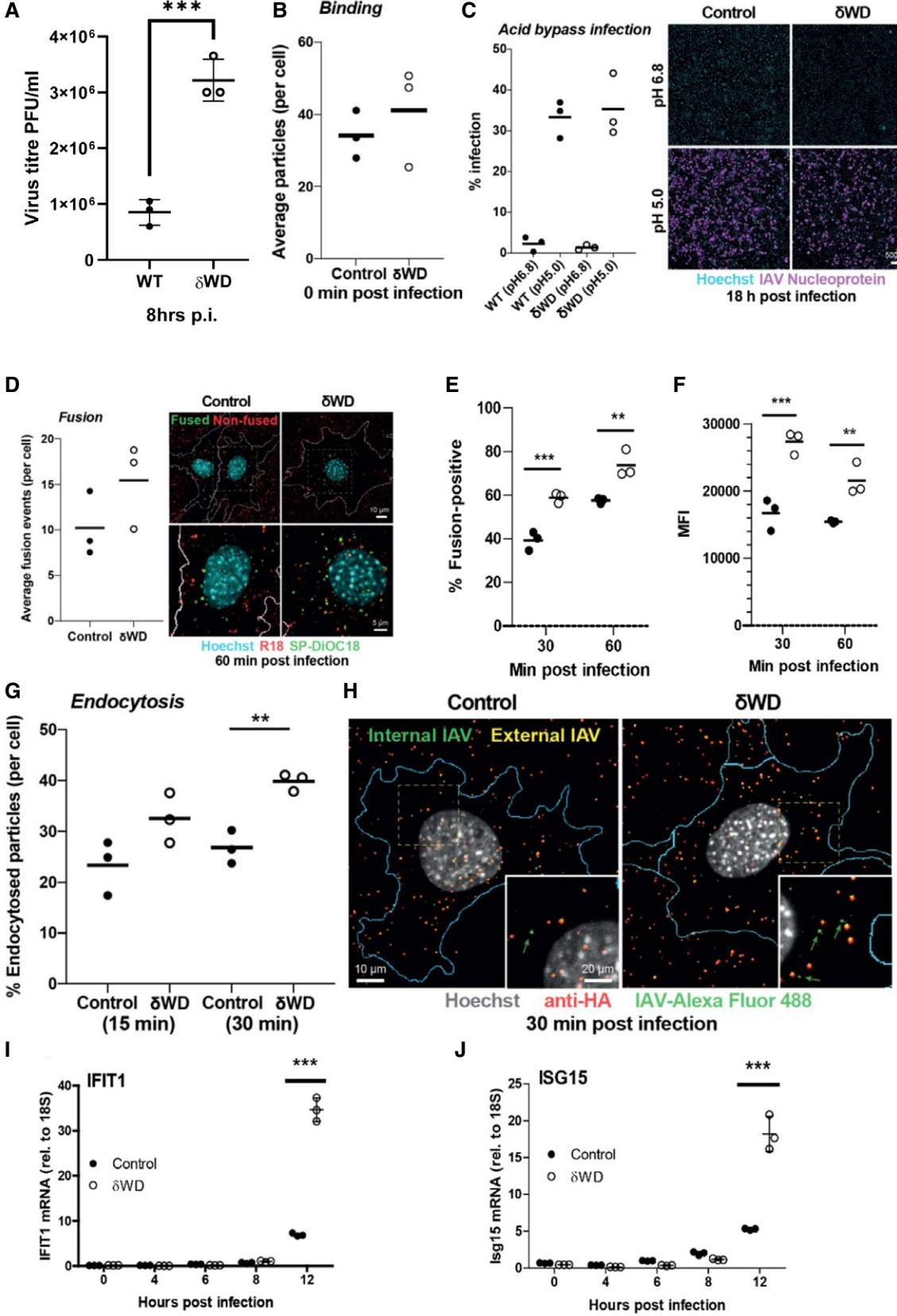

**Figure 6.**

◀

**Figure 6.  Non-canonical autophagy reduces virus replication, endosome fusion and interferon signalling.**

A   The WD domain of ATG16L1 reduces IAV replication in MEFs. Cells were infected with IAV, and virus titres were estimated at 8 h post-infection by plaque assay. Graphs show three individual experiments with a bar (± SD) at the mean and were compared using two-way ANOVA with Bonferroni post-tests ***$P < 0.001$).

B   IAV binding is similar between control and δWD MEFs. Cells were bound with IAV X31 labelled with AF488 on ice for 1 h and fixed immediately. The cells were stained with anti-HA (H3SKE, red) without permeabilisation to visualise extracellular IAV particles and counterstained with Hoechst (nuclei) and WGA-AF647 to label the cell membrane. Cells were imaged by automated confocal microscopy using a 40x objective, and the maximum intensity projection images were analysed to quantify the number of bound IAV particles per cell. The results and means (horizontal line) of $n = 3$ independent experiments (50-100 cells per experiment) are shown.

C   IAV infection following acid bypass is comparable between control and δWD MEFs. IAV was bound on ice for 1 h and warmed at either neutral pH (6.8) or low pH (pH 5.0) for 2 min, followed by incubation in STOP medium containing $NH_4Cl$ for 18 h. Cells were fixed and stained for NP and Hoechst, imaged by automated confocal microscopy and maximum intensity projection images analysed to quantify viral infectivity. The results from N = 3 independent experiments (5,000-10,000 cells quantified per experiment), and the means are shown. Scale bar; 500 μm.

D   IAV fusion with endosomes is increased in δWD MEFs. Cells were infected with dual-labelled (R18/SP-DiOC18) IAV for 1 h, fixed and counterstained with Hoechst (nuclei blue) and WGA-AF647 (cell perimeter in white). Cells were imaged by automated confocal microscopy, and the number of fusion events per cell (represented by number of SP-DiOC18 puncta) was quantified. Fused and non-fused viral particles are shown as green and red spots, respectively. The results and means (horizontal lines) of $n = 3$ independent experiments (30-60 cells per experiment) are shown. Representative cells overlayed with the cell boundary segmented from the WGA staining are shown on the right.

E, F   Numbers of fusion-positive cells are increased for δWD MEFs. MEFs from δWD (white circles) or littermate control mice (black circles) were incubated with dual-labelled (SP-DiOC18/R18) IAV at 4°C for 45 min and warmed to 37°C for 30 ad 60 min. Cells were harvested by trypsinisation, fixed in PFA and analysed by flow cytometry to determine to determine percentage of cells positive for fusion (E) and median fluorescence intensity (MFI) of de-quenched SP-DiOC18 signa (F). Graphs show individual replicates ($n = 3$) with a horizontal line at the mean and were compared using two-way ANOVA with Bonferroni post-tests **$P < 0.01$, ***$P < 0.001$). pH-dependent fusion was assessed by adding bafilomycin A1 to the infection assay (Fig EV5).

G, H   Endocytosis of IAV increases in δWD MEFs. MEFs from δWD or littermate control mice were incubated with Alexa Fluor 488-labelled IAV at 4°C for 1hr and warmed to 37°C for 15. Cells were fixed and permeabilised and stained with antibody against HA (red). The blue line indicates the plasma membrane of the cell. Green arrows indicate examples of viruses only labelled with green fluorescence. Double-labelled virus particles (green-red) represent non-endocytosed particles bound at the cell surface, whereas green particles represent endocytosed particles. The results and means of $n = 3$ independent experiments (50-100 cells per experiment) are shown. Student's *t*-test; $P < 0.01$. The dotted square in the top row panels is enlarged in the bottom row. Scale bars; 10 μm (top row) and 5 μm (bottom row).

I, J   Interferon signalling increases in δWD MEFs. MEFs from δWD or littermate control mice were infected with IAV. At the indicated time points, mRNA transcripts were evaluated by qPCR for IFIT1 (I) and ISG15 (J). Data from 3 replicates are shown, and bars represent the mean ± SD and were compared by two-way ANOVA with Bonferroni post-tests (***$P < 0.001$)

spread of respiratory viruses underlines an urgent need to understand how the airways defend against viral infection. In this study, we have analysed the role played by non-canonical autophagy in defending the respiratory tract against infection by IAV *in vivo*. Mice with systemic loss of non-canonical autophagy (δWD) showed profound sensitivity to infection by a low-pathogenicity murine-adapted IAV (A/X31) leading to extensive viral replication throughout the lungs, dysregulated cytokine production and fulminant pneumonia leading to high mortality and death usually seen after infection with virulent strains (Belser *et al*, 2011). These signs mirror the cytokine storms and mortality seen in humans infected with highly pathogenic strains of IAV such as the 1918 "Spanish" Influenza (Belser *et al*, 2011).

The observation that bone marrow transfers from wild-type mice were unable to protect δWD mice from IAV suggested that protection against IAV infection *in vivo* was independent of leucocytes and did not require non-canonical autophagy in leucocyte populations (e.g. macrophages, dendritic cells, neutrophils, granulocytes, lymphocytes). In a reciprocal experiment, the linker and WD domains of ATG16L1 were deleted specifically from myeloid cells. These mice, which lack non-canonical autophagy in phagocytic cells (LAP), but maintain non-canonical autophagy in other tissues, failed to show increased sensitivity to IAV infection. Thus, protection against severe IAV-associated disease in the respiratory tract of the host relies heavily on non-canonical autophagy in non-leucocyte populations.

Activation of non-canonical autophagy in phagocytic cells leads to LC3-associated phagocytosis (LAP) where TLR signalling and reactive oxygen species (ROS) recruit LC3 to phagosomes. A lack of

involvement of non-canonical autophagy/LAP in protection against IAV disease *in vivo* was surprising because the activation of LAP in phagocytic cells such as macrophages, dendritic cells and neutrophils would provide a powerful means of recognising and controlling microbial infection *in vivo*. *In vitro* studies show that activation of acid sphingomyelinase by *Listeria monocytogenes* (Gluschko *et al*, 2018) and subsequent ROS production by NOX2 recruit LC3 to phagosomes. Similarly, activation of TLR2 and NOX2 by *Legionella dumoffii in vitro* signals ULK1-independent translocation of LC3 to single-membraned vacuoles containing Legionella (Hubber *et al*, 2017). In both cases, LC3 promotes fusion with lysosomes. The observation that virulence factors such as the GP63 metalloprotease of *Leishmania major* and melanin of *Aspergillus fumigatus* prevent recruitment of NOX2 to phagosomes to prevent LAP (Akoumianaki *et al*, 2016; Matte *et al*, 2016; Kyrmizi *et al*, 2018) further suggests that non-canonical autophagy in phagocytes should provide a defence against infection. One reason for the discrepancy may be that the studies cited above have focused on *in vitro* experiments using microbes with a tropism for macrophages, rather than *in vivo* studies where pathogens encounter epithelial barriers.

Intranasal infection of mice with IAV results in rapid infection of principally airway and pulmonary epithelial cells (Akram *et al*, 2018). The results of *in vivo* challenge of radiation chimaeras and δWD[phag] mice strongly suggest that non-canonical autophagy in the epithelium rather than leucocytes is responsible for restricting IAV infection. This was supported by *ex vivo* experiments where virus titres and interferon responses were five- to 10-fold greater in precision-cut lung slices and MEFs from δWD mice. Furthermore, loss of non-canonical autophagy increased fusion of IAV envelope with

endosomes and increased activation of interferon signalling pathways. Both assays suggest that non-canonical autophagy reduces IAV entry and delivery of viral RNA to the cytoplasm. This would explain reduced interferon signalling, and at the same time the delayed escape of IAV into the cytoplasm would increase the transfer of endocytosed virus to lysosomes for degradation. The precise mechanisms employed by non-canonical autophagy to reduce virus entry from endosomes are unknown. This may involve recruitment of LC3 to endosomes by TMEM59 (Boada-Romero *et al*, 2016) to increase fusion with lysosomes, or by maintaining membrane repair during virus entry, as observed for bacteria such as *S.* Typhimurium and *Listeria monocytogenes* (Kreibich *et al*, 2015; Tan *et al*, 2018). A p22$^{phox}$-NOX2 pathway that recruits LC3 to vacuoles containing *S.* Typhimurium in epithelial cells (Huang *et al*, 2009) may also be activated during IAV entry and hamper lethal infection.

δWD mice infected with IAV appeared to be unable to resolve inflammatory responses resulting in sustained expression of pro-inflammatory cytokines, morbidity and a striking lung changes characterised by profuse migration of neutrophils into the airway at day 3 followed by macrophages on day 7. pDCs detect IAV-infected cells and produce large amounts of cytokines, in particular IFNαβ, that in severe infections can enhance disease (Davidson *et al*, 2014). The fact that morbidity in δWD mice could be decreased by depleting pDCs indicates that excessive cytokine production, amplified by pDCs, was a major factor. This is not due to a lack of non-canonical autophagy/LAP in pDC as bone marrow chimaeras of δWD mice with wild-type leucocytes have the same phenotype as δWD mice. IAV is recognised by endosomal TLR3 in respiratory epithelial cells and RIG-I detects virus replicating in the cytosol leading to activation of IRF3 and NFkB with subsequent induction of interferon, ISG and pro-inflammatory cytokine production (Iwasaki & Pillai, 2014). Increased inflammation may result directly from increased virus in the lungs, but the increased fusion of IAV envelope with endosomes in δWD mice may increase delivery of viral RNA to the cytoplasm resulting in the sustained pro-inflammatory cytokine signalling. A similar pro-inflammatory phenotype resulting from decreased trafficking of inflammatory cargoes is observed following disruption of non-canonical autophagy by LysMcre-mediated loss of Rubicon from macrophages or microglia (Martinez *et al*, 2016; Heckmann *et al*, 2019). Studies also show that the WD domain of ATG16L1 can modulate endocytosis of cytokine receptors (Serramito-Gómez *et al*, 2020). Interaction of the WD domain with the IL10 receptor (IL10-R), for example, promotes formation of endosomes containing IL-10/IL-10 receptor complexes leading to an enhanced anti-inflammatory signalling, that would be lost in δWD mice. Impaired recycling of Toll-like receptor 4, CD36 and the β-amyloid receptor TREM2 is also observed in microglia lacking the WD domain of ATG16L1 leading to neuroinflammation (Heckmann *et al*, 2020). Inhibition of receptor recycling results from slowed return of receptors to the plasma membrane, rather than increased endocytosis (Heckmann *et al*, 2019). This makes it unlikely that the increased fusion of IAV with endosomes we see in δWD cells results from upregulation of endocytosis following loss of the ATG16L1 WD domain.

We have dissected the roles played by conventional autophagy and non-canonical autophagy *in vivo* by removing the linker and WD domain from ATG16L1 to prevent conjugation of LC3 to single-membraned endo-lysosome compartments (Rai *et al*, 2019). Quantitative analysis of conventional autophagy by fluorescence microscopy of LC3 puncta (Fig 1B) and Western blot of LC3II (Fig 1C&D) did not reveal a loss of canonical autophagy in cells from δWD mice compared to controls. Nevertheless, it is not possible to exclude the possibility that removal of the WD and linker domains of ATG16L1 has a minor effect on canonical autophagy that might affect infection "in vivo". There are however examples where the WD domain of ATG16L1 has roles during infection that are separate from conventional autophagy, and this may be true also for control of IAV. The WD domain of ATG16L1 maintains membrane repair during Listeria infection independently of conventional autophagy (Tan *et al*, 2018). The Salmonella T3SS effector protein SopF reduces recruitment of LC3 to vacuoles containing *S.* Typhimurium by inhibiting the interaction between the WD domain of ATG16L1 and the vacuolar ATPase recruited to sites of vacuole damage (Xu *et al*, 2019). This promotes growth and virulence of *S.* Typhimurium, but is independent of FIP200, an essential component of conventional autophagy. An alternative approach to studying non-canonical autophagy "in vivo" has been to target pathways upstream of LC3 conjugation where deletion of Rubicon produces a selective block in LAP (Martinez *et al*, 2015; Heckmann *et al*, 2019). Rubicon stabilises the PHOX:NOX2 complex (Yang *et al*, 2009) allowing reactive oxygen species (ROS) to induce binding of ATG16L1 to endo-lysosome membranes (Martinez *et al*, 2015). Mouse models relying on loss of Rubicon show defects in the clearance of bacterial and fungal pathogens and apoptotic cells (Martinez *et al*, 2015; Martinez *et al*, 2016), but have not yet been studied in the context of viral infection. Furthermore, disruption of Rubicon leads to upregulation of IL-1β, IL6 and TNF-α secretion, and the mice fail to gain weight and develop an autoimmune disease that resembles systemic lupus erythematosus (Martinez *et al*, 2016; Heckmann *et al*, 2017). This exaggerated inflammation might make it difficult to predict if any altered responses to infection observed in Rubicon-/- mice, particularly lung inflammation, resulted directly from loss of non-canonical autophagy, or from upstream changes in cytokine regulation caused by loss of Rubicon.

Several non-canonical pathways leading to recruitment of LC3 to endo-lysosomal compartments, rather than phagosomes, are beginning to emerge. Non-canonical autophagy in microglia facilitates endocytosis of β-amyloid and TLR receptors to reduce β-amyloid deposition and inflammation in mouse models of Alzheimer's disease (Heckmann *et al*, 2019). This may involve interaction between the WD domain and TMEM59 which is required for β-amyloid glycosylation (Ullrich *et al*, 2010). Lysosomotropic drugs, which stimulate direct recruitment of LC3 to endosomes, create pH and osmotic changes that may mimic the consequences of viral infections that perturb endosome membranes or deliver viroporins to endo-lysosome compartments. It will be interesting to see if the WD and linker domains of ATG16L1 limit infection by other microbes at epithelial barriers *in vivo*, particularly infection of the respiratory tract by SARS-CoV-2. This may be true for picornaviruses where LC3 is recruited to enlarged endosomes during entry of foot-and-mouth disease virus (Berryman *et al*, 2012) and following LC3 accumulation on megaphagosomes in pancreatic acinar cells during coxsackievirus B3 infection (Kemball *et al*, 2010). In the specific cases of IAV and SARS-CoV-2, non-canonical autophagy at epithelial barriers is likely important for innate control of new pathogenic strains, where acquired immunity from previous infection may be absent or less effective. It will be valuable to assess

whether human allelic variants of ATG16L1 confer altered resistance/susceptibility to respiratory infections such as IAV and whether drug-based manipulation of non-canonical autophagy can increase resistance at the respiratory epithelial barrier.

# Materials and Methods

### Cell culture and virus

Influenza virus A/HKx31 (X31, H3N2) was propagated in the allantoic cavity of 9-day-old embryonated chicken eggs at 35°C for 72 h. Titres were determined by plaque assay using MDCK cells with an Avicel overlay.

### Mice

All experiments were performed in accordance with UK Home Office guidelines and under the UK Animals (Scientific procedures) Act1986.

The generation of δWD mice ($Atg16L1^{\delta WD/\delta WD}$) has been described previously (Rai *et al*, 2019) where they are called "E230" mice to distinguish them from E226 mice described in the same paper which lack both canonical and conventional autophagy. Generation of $\delta WD^{phag}$ and $Atg16L1^{fl/fl}$-LysMCre mice is described in detail in Appendix Fig S7. Comparisons were made using age- and sex-matched littermate control mice for each individual genotype. Generation and breeding of mice was approved by the University of East Anglia Animal Welfare and Ethical Review Body and performed under UK Home Office Project License 70/8232.

Influenza infection studies were performed at the University of Liverpool, approved by the University of Liverpool Animal Welfare and Ethical Review Body and performed under UK Home Office Project License 70/8599. Studies used 2- to 3-month-old male and female mice that had been back-crossed to C57BL/6J. Mice were maintained under specific pathogen-free barrier conditions in individually ventilated cages (Greenline GM500, Tecniplast) at a temperature of 22°C (± 2°C), humidity 55% (± 10%), light/dark cycle 12/12 h (7 am to 7 pm), food CRM(P) and RO or filtered water *ad lib*. Colonies were screened using the Charles River surveillance plus PRIA health screening profile every 3 months to ensure SPF status.

For IAV infection, animals were randomly assigned into multiple cohorts, anaesthetised lightly by the i.m. route with 150 mg/kg ketamine (Ketavet, Zoetis UK Ltd) and separate cohorts inoculated intranasally with $10^3$ PFU IAV strain X31 in 50 µl sterile PBS. Mice were infected between 9 and 11 AM. Animals were sacrificed at variable time points after infection by cervical dislocation. Tissues were removed immediately for downstream processing. Sample sizes of $n = 6$ were used as determined using power calculations and previous experience of experimental infection with these viruses. For survival analysis, a humane endpoint was determined using a scoring matrix that included excessive (>20%) weight loss.

To specifically deplete plasmacytoid dendritic cells (pDCs), mice were treated with anti-PDCA-1 (Cambridge Bioscience) or IgG2b isotype-matched control, using a dose of 500 mg per 200 ml via the i.p. route on day 1 of infection with IAV and every 48 h thereafter (Davidson *et al*, 2014).

### Generation and analysis of radiation chimaeras

The general strategy is shown in Fig EV3. Mice were subjected to whole body irradiation with 11 Gy in two doses 4 h apart using a $^{137}$Cs source in a rotating closed chamber. Bone marrow was collected from male wild-type C57BL/6-Ly5.1 (B6.SJL-*Ptprc$^a$Pepc$^b$*/ BoyCrl; Atg16L1$^{+/+}$) mice that are congenic for the CD45.1 allele or from δWD mice (that are congenic for CD45.2). The C57BL/6 CD45.1 marrows were used to enable confirmation of chimaerism by FACS analysis of bone marrow-derived cells as littermate control and δWD mice are CD45.2 (Fig EV3B). The femur and tibia of the donor mouse was collected and sterilised for 2 min in 70% ethanol. The ends of the bones were removed, and PBS was used to flush out the bone marrow through a 40 µm cell sieve. Red blood cell lysis was performed using 0.83% ammonium chloride, and the cells were washed twice in PBS and resuspended at a concentration of $10^7$ cells/ml. T-cell depletion was performed prior to transfusion by using a commercial mouse hematopoietic progenitor cell isolation kit (EasySep, STEMCELL™ Technologies, #19856).

After depletion, $10^6$ donor bone marrow cells were injected into each irradiated mouse by tail vein injection 3 h following irradiation. Mice were then allowed to recover for 12 weeks with daily monitoring of mouse weights and general condition for at least the first two weeks to monitor for any severe radiation sickness or illness due to being immunocompromised.

For chimaerism analysis, approximately $10^6$ spleen cells were analysed by flow cytometry using fluorochrome-conjugated monoclonal antibodies specific for CD45.1 (clone A20 eBioscience) and CD45.2 (clone 104 eBioscience). As shown in Fig EV3B, in the groups where CD45.1 marrow was transplanted, all mice were > 95% chimaeric.

### Flow cytometric analysis of cells

Bronchoalveolar lavage (BAL) fluid was obtained by lavage of mice via the trachea using 1 ml ice-cold RPMI containing 5% FCS. For lung tissue, single-cell suspensions were made from minced lung and subjected to collagenase and DNase I digestion and then treated with ACK buffer to remove red blood cells. In both cases, approximately $10^6$ cells were incubated in 100 µl of Fc block (clone 2.4G2, BD Biosciences) diluted in PBS, 2% FCS (PBS-FCS) for 15 min at 4°C prior to the addition of fluorochrome-conjugated monoclonal antibodies and incubation for 30 min at 4°C in the dark. Cells were then washed in PBS-FCS, fixed in 4% paraformaldehyde in PBS for 15 min at 20°C prior to analysis on a MACSQuant Analyzer 10 (Miltenyi Biotec UK). Data were analysed using FlowJo (FlowJo, LLC). Antibodies used included CD45, Ly6G, CD11c, CD11b and F4/80 (all eBioscience). Neutrophil populations in BAL were identified as CD45$^+$, CD11c$^-$, CD11b$^+$ and Ly6G$^+$. Macrophage/monocyte populations in lung tissue were identified as CD45$^+$, CD11c$^-$, CD11b$^+$ and F4/80$^+$.

### Histology, immunohistochemistry

Tissues were fixed in 4% buffered paraformaldehyde (PFA; pH7.4) for 24 h and routinely paraffin wax embedded. Consecutive sections (3-5 µm) were either stained with haematoxylin and eosin (HE) or used for immunohistochemistry (IH).

IH was performed to detect influenza antigens and to identify neutrophils and neutrophil extracellular traps (NETs) and macrophages using the horseradish peroxidase (HRP) and the avidin–biotin complex (ABC) method. The following primary antibodies were applied: goat anti-IAV (Meridian Life Sciences Inc., B65141G), rat anti-mouse Ly6G (clone 1A8, BioLegend; neutrophil marker), rabbit anti-Iba-1 (antigen: AIF1; Wako Chemicals; microglia/macrophage specific marker) and rabbit anti-histone H3 (citrulline R2 + R8 + R17; Abcam; NET marker). Briefly, after deparaffinization, sections underwent antigen retrieval in citrate buffer (pH 6.0, 20 min at 98°C) followed by blocking of endogenous peroxidase (peroxidase block, S2023, Dako) for 10 min at room temperature (RT). Slides were then incubated with the primary antibodies (diluted in dilution buffer, Dako) for a) Iba-1 (60 min at RT), followed by a 30 min incubation at room temperature with the secondary antibody (Envision mouse and rabbit, respectively, Dako) in an autostainer (Dako), and b) Ly6G (60 min at RT), followed by rabbit anti-rat IgG and the ABC kit (both 30 min at RT; Ventana). Staining for histone H3 was undertaken with an autostainer (Discovery XT, Ventana), using citrate buffer, dilution buffer and detection kits provided by the manufacturer. The antibody reaction was visualised with 3,3'-diaminobenzidine, and sections were counterstained with haematoxylin.

### Statistical analysis

Data were analysed using the Prism package (version 5.04 GraphPad Software). $P$ values were set at 95% confidence interval. A repeated-measures two-way ANOVA (Bonferroni post-test) was used for time-courses of weight loss; two-way ANOVA (Bonferroni post-test) was used for other time-courses; log-rank (Mantel–Cox) test was used for survival curves; one-way ANOVA (Tukey's post hoc) was used to compare three or more groups side-by-side; Mann–Whitney $U$-test was used to compare two groups. Numbers of replicates are shown in the individual figure legends. All differences not specifically stated to be significant were not significant ($P > 0.05$). For all figures, $*P < 0.05$, $**P < 0.01$, $***P < 0.001$, and $****P < 0.0001$.

### Primary cell culture

Mouse embryonic fibroblasts (MEFs) were generated by serial passage of cells taken from mice at embryonic day 13.5 and cultured in DMEM (Thermo Fisher Scientific, 11570586) with 10% FCS. Bone marrow-derived macrophages (BMDMs) were generated from femur and tibia flushed with RPMI-1640 (Sigma, R8758). Macrophages were generated from adherent cells in RPMI-1640 containing 10% FCS and M-CSF (Peprotech, 315-02) (30 ng/ml) for 6 d. Macrophage populations were quantified by FACS using antibodies against CD16/CD32, F4/80 and CD11b (BioLegend, 101320, 123107).

### Precision-cut lung slices

Infection of *ex vivo* lung slices was used to examine the responses of lungs without any contribution from recruited leucocytes, which could not be present. Mouse lungs were inflated with 2% low melting point agarose in HBSS and then sliced into 300 μm sections using a vibrating microtome. They were then cultured overnight in DMEM/F12 medium (Thermo Fisher 21331020) prior to infection with IAV.

### IAV binding and endocytosis assay

IAV X31 was labelled with Alexa Fluor 488 (Thermo Fisher Scientific) as described (Hoffmann *et al*, 2018). IAV entry assays were performed as previously described (Banerjee *et al*, 2013; Banerjee *et al*, 2014; Miyake *et al*, 2019). Sub-confluent monolayers of MEFs cultured on optical 96-well plates (Greiner 655090) in DMEM 10% FBS were incubated with the labelled virus on ice for 1 h in infection medium (DMEM, 50 mM HEPES pH 6.8, 0.2% BSA) and warmed for 0 (for binding experiments) 15 and 30 min at 37°C by addition of warmed infection medium and transfer of the plates to 37°C, followed by fixation with 4% paraformaldehyde in PBS. Fixed cells were blocked for 1 h with 1% BSA in PBS before incubation with a mouse anti-haemagglutinin monoclonal antibody (H3SKE) for 1h at RT, followed by a 30-min incubation with a second goat anti-mouse-Alexa Fluor 594 antibody (Thermo Fisher Scientific, A11005), together with Hoechst 33342 (Thermo Fisher Scientific) and Wheat Germ agglutinin labelled with Alexa Fluor 647 (WGA-AF647, Thermo Fisher Scientific, W32466). Image acquisition was performed with a Yokogawa CQ1 spinning disc confocal microscope, using a 40x air objective with an NA of 0.95, in 4-channel mode. Image analysis was performed using the Cell Path Finder version 3.04.03.02 (Yokogawa Electric Corporation). Briefly, images were subjected to segmentation analysis, wherein overlap between red (H3SKE/goat anti-mouse-AF594) and green (X31-AF488) signals was quantified within the boundaries of a segmented cell outline generated by the same programme, thus allowing the quantification of endocytosed virus per cell. Signals colocalising with the nucleus (Hoechst) were omitted. Double-labelled virus particles (Green-Red) represent non-endocytosed particles bound at the cell surface, whereas green particles represent endocytosed particles. Within cell outlines, the number of green (endocytosed) particles normalised against the total number of particles (surface-bound and endocytosed, Green-Red + Green) gives the fraction of endocytosed particles.

### IAV endosome fusion assessed in cell populations by FACS

The envelope of purified IAV (0.1 mg protein mL$^{-1}$) was labelled using an ethanol solution containing 33 μM 3,3'-dioctadecyloxacarbocyanine (DIOC18) and 67 μM octadecyl rhodamine B (R18). Aggregated virus was removed by a 0.22 μm filter (Millipore). Sub-confluent monolayers of MEFs cultured in DMEM (50 mM HEPES, 0.2% BSA) were incubated with the labelled virus at 40°C for 45 min and warmed to 37°C for increasing times. Cells were harvested by trypsinisation and fixed in 4% PFA for 20 min. Cell pellets (2500 rpm, 4 min) were resuspended in 100 μL FACS buffer (1xPBS 1%BSA) and analysed by FACs using a NovoCyte Flow cytometer FlowJo software. pH independent fusion was assessed by adding bafilomycin A1 to the infection assay.

### IAV fusion assessed in single cells by microscopy

X31/R18 double-labelled X31 was prepared as herein described. For the fusion experiment, MEF cells were incubated in binding media

for 1h on ice and then warmed for 1h at 37°C by addition of warm binding media and transfer of the plates to 37°C, followed by fixation with 4% paraformaldehyde in PBS. The cells were then labelled with Hoechst and WGA-AF647 as herein described. Image acquisition was performed with a Yokogawa CQ1 spinning disc confocal microscope, using a 40x air objective with an NA of 0.95, in 4-channel mode. Image analysis was performed using the Cell Path Finder version 3.04.03.02 (Yokogawa Electric Corporation).

### IAV acid bypass assay

Sub-confluent monolayers of MEFs cultured on optical 96-well plates (Greiner 655090) in DMEM 10% FBS were incubated with IAV X31 on ice for 1 h in infection medium (DMEM, 50 mM HEPES pH 6.8, 0.2% BSA) and warmed for 2 min at 37°C by addition of warm medium (pH 6.8) or warm medium adjusted to low pH (pH 5.0) by citrate buffer, on a metal block immersed in a 37°C water bath. Another warm metal block was placed on top of the plate. The medium was replaced with STOP medium (DMEM, 50mM HEPES pH7.4, 20 mM $NH_4Cl$) to block viral entry through the endocytic route, and the plate was transferred to 37°C and incubated for 18 h, fixed with 4% paraformaldehyde in PBS. Fixed cells were blocked for 1 h with 1% BSA in PBS before incubation with a mouse anti-nucleoprotein monoclonal antibody (HB-65, ATCC) for 1h at RT, followed by a 30-min incubation with a second goat anti-mouse-Alexa Fluor 647 antibody (Invitrogen, A21235), together with Hoechst 33342 (Thermo Fisher Scientific). Image acquisition was performed with a Yokogawa CQ1 spinning disc confocal microscope, using a 10x air objective with an NA of 0.40, in 2-channel mode. Image analysis was performed using the Cell Path Finder version 3.04.03.02 (Yokogawa Electric Corporation).

### Image acquisition by spinning disc confocal microscopy

Image acquisition was performed with a Yokogawa CQ1 spinning disc confocal microscope, using a 10x air objective with an NA of 0.40, or a 40x air objective with an NA of 0.95. Imaging was performed with up to four excitation laser lines (405/488/561/640nms) with spinning disc confocal. For 10x objective, images were acquired with 5 z-stacks to cover 40 µm; for 40x objective, images were acquired with 20 z-stacks to cover 30 µm. Maximum intensity projected images were used for image analysis using the Cell Path Finder version 3.04.03.02 (Yokogawa Electric Corporation).

### Autophagy and non-canonical autophagy

Autophagy was activated by incubating MEF cells in Hanks balanced salt solution (HBSS) (Thermo Fisher, 11550456) for 2 h at 37°C. Non-canonical autophagy was stimulated in MEFs with monensin (Sigma-Aldrich, M5273) or chloroquine (Sigma-Aldrich, C6628) with a final concentration of 100 µM for 2 h. Numbers of fluorescent LC3 puncta were quantified by fluorescence microscopy using spot function software (IMARIS package [BITPLANE sScientific Software]) to locate puncta ranging from 0.5–1.0 µm diameter. LC3 vacuoles were identified by eye as rings of fluorescence ranging between 2 and 8 µm diameter. Non-canonical autophagy/LAP was assessed in BMDMs by incubation with Zymosan A (Alexa Fluor 594-labelled; Thermo Fisher Z23374).

Phagosomes containing Zymosan were observed by eye using immunofluorescence microscopy. Line profile analysis was applied across the centre of the phagosome using ImageJ. The extent of recruitment of LC3 to the phagosome was calculated by subtracting the LC3 pixel intensity at the centre of the phagosome from intensity observed at the perimeter. Conjugation of LC3 to PE was also assessed by Western blot from the ratio of LC3II.

### qPCR for cytokine transcription

Lung lobes were snap-frozen and homogenised using a TissueLyser (Qiagen). Tissue culture cells were washed twice using PBS. Total RNA was extracted by TRIzol–chloroform (Thermo Fisher 15596018) and purified by RNeasy MinElute Cleanup Kit (Qiagen 74204). RNA was analysed by qPCR using SYBR Green/7500 (Thermo Fisher S7563) Standard Real-Time PCR System (Applied Biosystems, Grand Island, NY) and primer sets as detailed in Table S1. Relative amounts of mRNA expression were normalised to 18S rRNA (Table 1).

### Western blotting

Cells were lysed using M-PER reagent (Thermo Fisher 78501) with complete protease inhibitor cocktail (Sigma, 04693159001) and clarified by centrifugation. Extracted proteins (20 µg) were separated on a precast 4–12% gradient SDS–PAGE gels (Expedeon, NBT41212), transferred to immobilon PVDF (Millipore, IPFL00010) and probed using antibodies for ATG16L1 (MBL M150-3), LC3A/B (Cell Signaling 41085) and actin (Sigma, A5441). Primary antibodies were detected using IRDye-labelled secondary antibodies (LI-COR biosciences, 926-32211, 926-68020) and visualised by Odyssey infrared system (LI-COR).

### Fluorescence imaging

Cells were fixed in ice-cold methanol, and non-specific binding was blocked using 5% goat serum plus 0.3% Triton X-100 in PBS followed by incubating with anti LC3A/B (Cell Signaling 4108) or anti-ATG16L1 (MBL M150-3). Cells were washed and then incubated with anti-rabbit-Alexa 488 (Thermo Fisher 10729174). After washing, cells were counterstained with 4', 6 diamidino-2-phenylindole (DAPI) (Thermo Fisher Scientific, 10116287) and mounted with Fluoromount-G (Cambridge Bioscience). Cells were imaged on a Zeiss Imager M2 Apotome microscope with a 63x, 1.4 NA oil-immersion objective.

**Table 1.  Primer sequences for mRNAs analysed by RT–qPCR**

| Target | Catalogue number[a] |
|---|---|
| ISG15 | QT00322749 |
| IFIT1 | QT01161286 |
| IL-1β | QT01048355 |
| TNF-α | QT00104006 |
| CXCL1 | QT00115647 |
| CCL2/MCP-1 | QT00167832 |
| 18S ribosomal RNA | QT02448075 |

[a]Catalogue numbers refer to validated QuantiTect primer sets (Qiagen).

## Data availability

This study includes no data deposited in external repositories.

**Expanded View** for this article is available online.

## Acknowledgements

This work was funded in part by Biotechnology and Biological Sciences Research Council (BBSRC) grant BB/R00904X/1 to JPS, JLC SRC, PPP, UM and TW, and through BBSRC Institute Strategic Programme Gut Microbes and Health BB/R012490/1: BBS/E/F/000PR10353, BBS/E/F/000PR10355. H3SKE was a kind gift of John Skehel. We thank Alina Rozanova with help on IAV Alexa Fluor labelling and IAV endocytosis assay set up, Mahomi Suzuki and Toshiaki Endo of Yokogawa Electric Corporation for help with image analysis.

## Author contributions

TW, JPS, PPP, SRC and JLC conceived the experiments. Mouse strains were generated by UM and genotyped by MJ, SR and WZ. Immunological homeostasis was assessed by WZ, AM and AZ. Animal infections were carried out by JPS, YW, WZ and PS and histology and immunohistology by AK. IAV fusion was analysed by YY with DB and BB. Downstream analysis was performed by YW, WZ, PS, TP and PPP. *In vitro* analysis was performed by BB, TP, YY and PPP. The manuscript was drafted by TW, JPS, YB, PPP, RAT and UM and edited and approved by all authors.

## Conflict of interest

The authors declare that they have no conflict of interest.

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
