## [Review Process File · The EMBO Journal]

Non-canonical autophagy functions of ATG16L1 in epithelial cells limit lethal infection by influenza A virus

Yingxue Wang, Parul Sharma, Matthew Jefferson, Weijiao Zhang, Ben Bone, Anja Kipar, David Bitto, Janine Coombes, Timothy Pearson, Angela Mann, Aleksandra Zhekova, Yongping Bao, Ralph Tripp, Yohei Yamauchi, Simon Carding, Ulrike Mayer, Penny Powell, James Stewart, and Thomas Wileman DOI: [10.15252/embj.2020105543](https://doi.org/10.15252/embj.2020105543)

Corresponding author(s): James Stewart (j.p.stewart@liv.ac.uk) , Thomas Wileman (t.wileman@uea.ac.uk)

Review Timeline:

Submission Date:	6th May 20
Editorial Decision:	3rd Jul 20
Revision Received:	30th Oct 20
Editorial Decision:	19th Nov 20
Revision Received:	23rd Dec 20
Accepted:	8th Jan 21

Editor: Elisabetta Argenzio

Transaction Report:

Thank you for submitting your manuscript entitled "The WD and linker domains of ATG16L1 required for non-canonical autophagy limit influenza A virus" [EMBOJ-2020-105543] to The EMBO Journal. Your study has been sent to three reviewers for evaluation, whose reports are enclosed below.

As you can see, the referees concur with us on the general interest of your study. However, they also raise several critical points that need to be addressed before they can support publication here. In particular, they request you to further investigate the fusion of IAV with endosomes in WD mutants of ATG16L1, the role of LAP through functional blockade of NOX and/or Rubicon and how deletion of the WD domain leads to an enhanced influenza-induced inflammatory response. Furthermore, the referee ask you to perform proper quantification and statistical analysis of the experiments.

Given the overall interest of your study, I am pleased to invite submission of a revised manuscript as indicated in the reports attached herein. I would like to point it out that addressing all referees' points in a conclusive manner, as well as a strong support from the referres, would be essential for publication in The EMBO Journal.

REFEREE REPORTS

Referee #1:

GENERAL COMMENTS

In this manuscript, Wang et al., investigated the role of non-canonical autophagy in influenza infection. Authors showed that mouse compromised for LC3 associated phagocytosis (LAP) are more sensitive to influenza infection and controls lung inflammation during IAF infection. They, furthermore, show that these functions of non-canonical autophagy are independent of phagocytosis and that non-canonical autophagy deficient mice were more prone to endosomal

acidification (which is required for delivery of viral ribonuclear proteins to enter the cytoplasm and subsequent replication. Overall, the conclusions are supported by the data and the overall quality of the data is good. Although, the role of WD mutant in IAV infection has been addressed previously (Fletcher et al., 2018; EMBO J), the present study illuminates further the role of ATG16L1 WD mutant in inflammation using animal model, which is informative and important.

1. Animal work

The authors were very thorough and made an excellent case supporting their hypothesis with some really complex mouse work. I think they leave no question that non-canonical autophagy, particularly in lung epithelial cells, is an essential component of innate defense against IAV. All of their ex-vivo, and in-vitro work support their story and mesh really well with the in-vivo data (weight, survival, and PFU's). The cytokine expression profiles, and histology/IHC data definitively help explain the high mortality of the δ WD mice. The different mouse crosses and radiation chimeras really helped pinpoint the cell and tissue types that were most important for clearance of IAV.

2. Figure 1 is not labelled properly, everything seems to be mixed. Authors should label figure panels properly (e.g. A for monensin, B for starvation, C for LC3 lipidation etc.).
3. Is the LC3 western blots shown in left (middle) corner of each panel from starved sample? Is there increased canonical autophagy in Fig. 1B (monensin treatment)?
4. Authors should quantify the autophagy (LC3 puncta/cell) in starved vs monensin treated cells in Fig. 1B, this would be important as this will explain if non canonical autophagy shifts to canonical autophagy in WD deleted mutant of ATG16L1.
5. Previous report suggests reduced lipidated LC3, which corresponds to LC3 puncta (membrane associated LC3) in monensin treated samples in WD mutant samples (Fletcher et al., 2018; EMBO J), so there should be relatively less overall LC3 puncta but it look more in this case. Authors need to clarify this.
6. What exactly is happening with monensin? I feel that this is an opportunity to go beyond just saying that monensin is an ionophore in past studies by similar groups. I leave up to the editor to decide how important this is, but I think the use of monensin must be better characterized in terms of its molecular and physiological effects.
7. Fig. 5: author's claim about noninvolvement of LAP seems weak; authors should either stay away from making these claims or provide some additional experimental data using NOX2 and/or Rubicon.
8. Fig. 6A : authors nicely showed more efficient fusion of IAV with endosomes in WD mutant, is there a way to prove using fluorescence/confocal microscopy?
9. It would also be good to present the data in a graphical form from \geq three independent experiments with appropriate statistics.

Referee #2:

In their manuscript, Wang et al investigate IAV infectivity in mice lacking the WD40 domains of ATG16L1 which were previously shown to be deficient in ATG8 lipidation on single membranes. The authors show that these mice exhibit reduced survival upon virus infection, higher virus titre and increased cytokine production at later time points after initial exposure to the virus. The authors show that this is not a result of defect in the myeloid lineage of WD40-deleted mice, but rather a defect in virus uptake when tested in MEF cells derived from these mice. Overall, this is an

interesting study. However, the conclusion that virus-endosome fusion is defective in the absence of the WD40 domains is not entirely convincing in this study. More specific comments are as follows:

Major comments:

Total deletion of the WD40 of ATG16L1 does not solely disrupt single-membrane lipidation of ATG8 proteins as this region also contains sequences required for recognition of bacterial infection (that may involve canonical autophagy) and the T300 variant region (dispensable for non-canonical autophagy). This is the main downside of using this mouse model to study ATG8 lipidation on single members and should be discussed in this manuscript.

Pg.4, line 21: WIPI2 binding is not located on the CCD of ATG16L1 but rather in the downstream region termed "linker" in the manuscript. This should also be corrected in Fig.1A and related legend as well as throughout the manuscript. The authors can refer to Dooley et al, 2014, Parkhouse et al, 2013 and Fujita et al, 2013 (as well as multiple other publications).

Fig.1B-C: it would be more informative for the authors to compare ATG16L1 levels from WT and delWD40 mice side by side to test any differences in expression. In addition, the blots showing LC3 lipidation during starvation lack any controls (e.g. untreated sample) making it not possible to judge whether the cells are responding to HBSS treatment. Comparing WT and delWD40 ATG16L1 cells side by side in this experiment would provide better evidence that canonical autophagy is not affected to support their conclusion.

The mouse model utilised in the manuscript is based on Rai et al which has generated multiple deletions in the WD40 domain of ATG16L1. The authors have not specified which one is being used in the manuscript (addition of ATG16L1 residue numbers that have been deleted would be important).

Fig.2D: the authors should also show IHC images of non-infected lungs to show difference in lung physiology under basal conditions. Are the apparent gross differences in lung histology also present in uninfected mice? Is this of significance to the phenotype observed?

In Fig.3, the authors show no difference in cytokine production at early time point of viral infection (up to 3 days). However, in Fig.6 they show that virus uptake is affected in the absence of WD40 domains. A discussion of why early cytokine production is not affected would be useful. The authors should also measure virus titre at various time points (Fig.2C) to confirm in vivo the findings using cultured MEF cells (Fig.6).

Fig.6: the differences in virus uptake in MEFs lacking WD40 should be confirmed using alternative assays. Could an increase in virus titre be a result of reduced lysosomal targeting and/or enhanced replication in the absence of non-canonical ATG8 lipidation? These findings require more detailed kinetic analyses to confirm the underlying mechanism contributing to enhanced virus titre.

Minor comments:

Pg.5, Lines 1-2: do the authors mean tissue-specific deletion of autophagy in mice exhibit pro-inflammatory phenotype? It would be good to clarify this and include the proper references.

Some figures have not been cited in the main text, including: Figures, 1A, S1, and S2.

Fig.1B-C: fluorescence images are saturated (especially in B).

Pg.6, Lines 12-13: Saitoh et al employed deletion of CCD (inhibitory of ATG16L1 known activities) rather than complete deletion of ATG16L1 as indicated in the manuscript.

Pg.8, Line 18: the authors conclude that deletion of ATG16L1 "prevented rapid weight loss". This conclusion is not clear in Fig.S3.

Fig.4C: labels are missing in this figure

Referee #3:

Wang and colleagues have investigated the role of the WD and linker domains of ATG16L1 in the host response to influenza A virus infection. To this end, they used mice lacking the WD and linker domains of ATG16L1, which are defective in non-canonical autophagy. They found that in the absence of non-canonical autophagy, mice are highly susceptible to influenza A infection, with cytokine storm, lung inflammation, and high mortality. Moreover, using phagocyte-specific deletion of the WD domain of ATG16L1, the authors provided evidence that susceptibility of influenza A infection was not mediated by phagocytes.

This study provides clear evidence that non-canonical autophagy is an essential component of the innate defence system that protects against influenza A infection and lethal inflammation in the lungs. This knowledge may certainly be applied to understand the pathology of other respiratory viruses.

There are some concerns that remain unanswered:

1- How does the loss of non-canonical autophagy/ATG16L1 WD domain lead to the triggering of inflammatory cytokine production?

2- What is the fate of LC3 during influenza A infection in mice lacking the WD domain of ATG16L1?

Referee #1:

Animal work

The authors were very thorough and made an excellent case supporting their hypothesis with some really complex mouse work. I think they leave no question that non-canonical autophagy, particularly in lung epithelial cells, is an essential component of innate defense against IAV. All of their ex-vivo, and in-vitro work support their story and mesh really well with the in-vivo data (weight, survival, and PFU's). The cytokine expression profiles, and histology/IHC data definitively help explain the high mortality of the δ WD mice. The different mouse crosses and radiation chimeras really helped pinpoint the cell and tissue types that were most important for clearance of IAV.

Q. Figure 1 is not labelled properly, everything seems to be mixed. Authors should label figure panels properly (e.g. A for monensin, B for starvation, C for LC3 lipidation etc.).

The figure has been reworked. Panels and labels have been put in place to increase clarity as requested.

Q. Authors should quantify the autophagy (LC3 puncta/cell) in starved vs monensin treated cells in Fig. 1B, this would be important as this will explain if non canonical autophagy shifts to canonical autophagy in WD deleted mutant of ATG16L1.

The generation of LC3 puncta after starvation and the recruitment of LC3 to vacuoles during non-canonical autophagy have been quantified using fluorescence microscopy and this is added as Figure 1B. We have extended the analysis to include chloroquine as a second means of inducing non-canonical autophagy. We did not see a shift to canonical autophagy in δ WD cells.

p6 In20. "Quantification of western blots (Fig 1D) showed that the LC3II signals in control and δ WD MEFs were similar after starvation suggesting that autophagy was equally active in the two cell types."

Q. Is the LC3 western blots shown in left (middle) corner of each panel from starved sample? Is there increased canonical autophagy in Fig. 1B (monensin treatment)?

The generation of LC3II after induction of autophagy or non-canonical autophagy has been quantified by western blot (Figure 1C and D). We have extended the analysis to include chloroquine as a second means of inducing non-canonical autophagy. We did not see increased canonical autophagy in the δ WD cells. We have provided an explanation for activation of canonical autophagy by lysosomotropic agents in both control and δ WD cells in the text as follows.

P6 In17. "Monensin and chloroquine raise lysosomal pH and the consequent inhibition of proteolysis slows the efflux of amino acids from the lysosome. This in turn inhibits the Ragulator-Rag:MTORC1 complex and induces autophagy. Previous work (Fletcher et al 2018) has shown that monensin activates conventional autophagy and at the same time raised lysosomal pH slows fusion of autophagosomes with lysosomes. This explains the accumulation of small LC3

puncta and increased LC3II observed in δ WD cells incubated with monensin or chloroquine. Quantification of western blots (Fig 1D) showed that the LC3II signals in control and δ WD MEFs were similar after starvation suggesting that autophagy was equally active in the two cell types. The LC3II signal in MEFs incubated with monensin or chloroquine was however lower than controls, which would be consistent with loss of non-canonical autophagy from δ WD cells”.

Q. Previous report suggests reduced lipidated LC3, which corresponds to LC3 puncta (membrane associated LC3) in monensin treated samples in WD mutant samples (Fletcher et al., 2018; EMBO J), so there should be relatively less overall LC3 puncta but it look more in this case. Authors need to clarify this.

Q. What exactly is happening with monensin? I feel that this is an opportunity to go beyond just saying that monensin is an ionophore in past studies by similar groups. I leave up to the editor to decide how important this is, but I think the use of monensin must be better characterized in terms of its molecular and physiological effects.

Monensin is an ionophore that raises lysosomal pH by exchanging protons for Na⁺ while chloroquine acts as a weak base to neutralise acid. Their ability to induce non-canonical autophagy is documented in Fletcher et al in EMBOJ. Rather than unravel possible off-pathway effects of monensin we have extended the study to include chloroquine in figure 1, and the results we obtain with chloroquine are the same as for monensin.

Q. Fig. 5: author's claim about noninvolvement of LAP seems weak; authors should either stay away from making these claims or provide some additional experimental data using NOX2 and/or Rubicon.

We have investigated the role played by NOX2 in non-phagocytic cells by observing the effects of inhibition of NOX2 by DPI on recruitment of LC3 to vacuoles during non-canonical autophagy in MEFs. This is added to as EV1C. This is described in the text.

P7 In5. “Studies in phagocytic cells have shown that non-canonical autophagy/LAP is downstream of Rubicon and PHOX:NOX2 ROS signaling (Martinez et al 2015). Addition of diphenylidonium (DPI), an inhibitor of NOX2, to MEFs incubated with monensin or chloroquine inhibited recruitment of LC3 to large vacuoles (Figure EV1C) indicating that WD domain-dependent non-canonical autophagy in non-phagocytic cells is also downstream of ROS signalling”.

We have also removed ‘LAP’ from the legend to figure 5 and from the text describing figure 5.

Fig. 5. Loss of non-canonical autophagy from phagocytes does not increase sensitivity to IAV infection.

Q. Fig. 6A : authors nicely showed more efficient fusion of IAV with endosomes in WD mutant, is there a way to prove using fluorescence/confocal microscopy?

Experiments on endocytosis have been extended to the use of automated confocal microscopy. These are presented in figure 6. They all show more efficient fusion with endosomes.

P13 In2. “The effect of non-canonical autophagy on IAV entry was tested using fluorescence de-quenching assay where the envelope of purified IAV was labelled with green (DiOC18) and red (R18) lipophilic dyes. Individual fusion events in cells

estimated by automated confocal microscopy (Fig 6D) shows that the number of fusion events per cell were increased almost two-fold after 60 minutes at 37⁰ in δ WD MEFs compared to control.”

P13In 10. “ Endocytosed viruses were also estimated by automated confocal microscopy (Fig. 6 G&H) of permeabilized cells and again there was increased endocytosis of IAV in δ WD MEFs at 30 minutes. Taken together the results showed that the WD domain of ATG16L1 slowed fusion of IAV with endosome membranes.”

Q. It would also be good to present the data in a graphical form from =/> three independent experiments with appropriate statistics.

The results are presented in graphs (Fig6 E,F,G) using three independent experiments.

Referee # 2. Major comments:

Total deletion of the WD40 of ATG16L1 does not solely disrupt single-membrane lipidation of ATG8 proteins as this region also contain sequences required for recognition of bacterial infection (that may involve canonical autophagy) and the T300 variant region (dispensable for non-canonical autophagy). This is the main downside of using this mouse model to study ATG8 lipidation on single members and should be discussed in this manuscript.

We have discussed the role of the WD domain and autophagy during recognition of bacteria.

P17 In21 “Quantitative analysis of conventional autophagy by fluorescence microscopy of LC3 puncta (Fig 1B) and western blot of LC3II (Fig 1C&D) did not reveal a statistically significant loss of canonical autophagy in cells from δ WD mice compared to controls. Nevertheless, it is not possible to exclude the possibility that removal of the WD and linker domains of ATG16L1 has a minor effect on conventional autophagy that might affect infection ‘in vivo’. There are however examples where the WD-domain of ATG16L1 has roles during infection that are separate from conventional autophagy, and this may be true also for control of IAV. The WD domain of ATG16L1 maintains membrane repair during Listeria infection independently of conventional autophagy (Tan et al 2018). The Salmonella T3SS effector protein SopF reduces recruitment of LC3 to vacuoles containing S. Typhimurium by inhibiting the interaction between the WD domain of ATG16L1 and the vacuolar ATPase recruited to sites of vacuole damage (Xu et al 2019). This promotes growth and virulence of S. Typhimurium, but is independent of FIP200, an essential component of conventional autophagy.”

Pg.4, line 21: WIPI2 binding is not located on the CCD of ATG16L1 but rather in the downstream region termed "linker" in the manuscript. This should also be corrected in Fig.1A and related legend as well as throughout the manuscript. The authors can refer to Dooley et al, 2014, Parkhouse et al, 2013 and Fujita et al, 2013 (as well as multiple other publications).

The text has been changed in the introduction as has the stick diagram in Figure EV1A

P4 In19. “The mice (δ WD) lack the WD and linker domains of ATG16L1 that are required for conjugation of LC3 to endo-lysosome membranes²¹ but express the N-terminal ATG5-binding domain and the coiled coil domain (CCD) and linker residues up to glutamate at position 230 (E230) of ATG16L1 that are required for WIPI2 binding and autophagy²³.”

Fig.1B-C: it would be more informative for the authors to compare ATG16L1 levels from WT

and delWD40 mice side by side to test any differences in expression. In addition, the blots showing LC3 lipidation during starvation lack any controls (e.g. untreated sample) making it not possible to judge whether the cells are responding to HBSS treatment. Comparing WT and delWD40 ATG16L1 cells side by side in this experiment would provide better evidence that canonical autophagy is not affected to support their conclusion.

A side by side comparison of ATG16L1 levels by western blot has been included in Fig 1C. The signal from the ATG16L1 lacking the WD domain in δ WD MEFs is lower than the signal from WT ATG16L1. This could be because there is less protein in δ WD cells, or because the antibody binds less strongly to the truncated ATG16L1. Importantly, further experiments included in the resubmission show that this lower signal is not reflected in compromised canonical autophagy. As suggested by the referee an experiment following LC3II lipidation in response to starvation is shown at the bottom of Fig 1C and graphically in Fig 1D. The cells respond equally to HBSS in ability to generate LC3II. Canonical autophagy has also been assessed by quantitative microscopy of LC3 puncta (Fig 1 B) and again there are no differences in ability to generate LC3 puncta between the control and δ WD cells. We have also shown in the Rai et al paper describing the δ WD mice that tissue levels of p62 are the same as control mice. This indicates that expression of the truncated δ WD ATG16L1 'in vivo' can activate canonical autophagy to clear autophagy cargo to a level equivalent to control mice. This allows the δ WD mice to grow normally and maintain tissue homeostasis.

The mouse model utilised in the manuscript is based on Rai et al which has generated multiple deletions in the WD40 domain of ATG16L1. The authors have not specified which one is being used in the manuscript (addition of ATG16L1 residue numbers that have been deleted would be important).

These have been added to the text.

P5 In19. “Panels A and B of figure EV1 shows the rationale for removing the WD and linker domains from ATG16L1 to generate mice (δ WD) with a specific loss of non-canonical autophagy (E230 mice described in Rai et al).

Fig.2D: the authors should also show IHC images of non-infected lungs to show difference in lung physiology under basal conditions. Are the apparent gross differences in lung histology also present in uninfected mice? Is this of significance to the phenotype observed?

Lung histology has been added as EV2.

P8 In9. “Lungs from control and δ WD mice did not show signs of inflammation before infection (Fig EV2).”

In Fig.3, the authors show no difference in cytokine production at early time point of viral infection (up to 3 days). However, in Fig.6 they show that virus uptake is affected in the absence of WD40 domains. A discussion of why early cytokine production is not affected would be useful. The authors should also measure virus titre at various time points (Fig.2C) to confirm in vivo the findings using cultured MEF cells (Fig.6).

Virus titre data has been added as Fig 2C

P8 In4. “Virus titre in the lungs of both mice increased with time (Fig. 2C) and increased weight loss in δ WD mice was associated with an approx. log increase in lung virus titre at 5 days post-infection (d.p.i).

A comparison between in vivo and in vitro findings is given

P12 In 6. “Virus titres in precision cut lung slices (Fig. 5D) increased over 3 days and similar to the kinetics seen in vivo, titres from δ WD mice rose to 10-fold greater than controls.”

Fig.6: the differences in virus uptake in MEFs lacking WD40 should be confirmed using alternative assays. Could an increase in virus titre be a result of reduced lysosomal targeting and/or enhanced replication in the absence of non-canonical ATG8 lipidation? These findings require more detailed kinetic analyses to confirm the underlying mechanism contributing to enhanced virus titre.

Experiments on endocytosis have been extended to the use of automated confocal microscopy. These are presented in figure 6 (B, D, G&H). They all show more efficient fusion with endosomes suggesting reduced lysosomal targeting.

P12 In17. “IAV binding was analysed using fluorescent virus and Fig 6B shows that binding was similar between control and δ WD MEFs suggesting that subsequent steps of endocytosis and viral fusion may be increased in δ WD MEFs.

P13 In2. “The effect of non-canonical autophagy on IAV entry was tested using fluorescence de-quenching assay where the envelope of purified IAV was labelled with green (DiOC18) and red (R18) lipophilic dyes. Individual fusion events in cells estimated by automated confocal microscopy (Fig 6D) shows that the number of fusion events per cell were increased almost two-fold after 60 minutes at 37^o in δ WD MEFs compared to control”

P13 In10. “Endocytosed viruses were also estimated by automated confocal microscopy (Fig. 6 G&H) of permeabilized cells and again there was increased endocytosis of IAV in δ WD MEFs at 30 minutes. Taken together the results showed that the WD domain of ATG16L1 slowed fusion of IAV with endosome membranes.

Q. Could an increase in virus titre be a result of reduced lysosomal targeting and/or enhanced replication in the absence of non-canonical ATG8 lipidation?

We have carried out an acid bypass experiment to see if enhanced replication occurred in the absence of non-canonical ATG8 lipidation. In this experiment the virus enters directly from the plasma membrane. The new data in Figure 6C shows that virus replication does not differ between δ WD and control MEFs following acid bypass. This suggests that replication itself is not influenced directly by the WD domain of ATG16L1 and that the increased replication in δ WD cells result from increased fusion with endosomes.

P12 In 19. “The possibility that the WD domain of ATG16L1 affected replication of IAV independently of endocytosis was tested using an acid bypass assay. IAV was bound to cells at 4^o and warmed to 37^o for 2 min at pH 6.8 (control) or pH 5 to induce direct fusion with the plasma membrane. When replication was assessed by staining for nuclear protein there was no difference between δ WD MEFs and control (Fig 6C). This made it unlikely that the WD domain has a direct role in facilitating IAV replication.

Minor comments:

Pg.5, Lines 1-2: do the authors mean tissue-specific deletion of autophagy in mice exhibit pro-inflammatory phenotype? It would be good to clarify this and include the proper references.

Some figures have not been cited in the main text, including: Figures, 1A, S1, and S2.

Fig.1B-C: fluorescence images are saturated (especially in B).

Microscopy and line profile analysis have been used to provide a quantitative analysis of recruitment of LC3 to phagosomes in bone marrow derived macrophages (BMDM) induced by Zymosan. These are presented in new datasets Fig 1 E-H and show a clear loss of recruitment in δ WD BMDM.

Pg.6, Lines 12-13: Saitoh et al employed deletion of CCD (inhibitory of ATG16L1 known activities) rather than complete deletion of ATG16L1 as indicated in the manuscript.

This has been corrected.

P10, In4. "Macrophages cultured from embryonic livers from mice with loss of the coiled coil domain of ATG16L1 are unable to activate canonical autophagy and secrete high levels of IL1- β ¹⁷."

Pg.8, Line 18: the authors conclude that deletion of ATG16L1 "prevented rapid weight loss". This conclusion is not clear in Fig.S3.

We have clarified the results.

P10 In8. "This has been reported to increase resistance to IAV infection¹⁶, and this was also observed in mice used in our study (Fig. S3) where LysMcre-mediated loss of Atg16L1 resulted in weight loss similar to controls (Fig S3A), and reduced overall virus titre (Fig S3B).

Fig.4C: labels are missing in this figure.

The figure 4C compares titres at 5dpi. The symbols used for the mouse genotype are shown in panels A and B. This has been added to the figure legend.

P25." Data for individual animals are shown using symbols described in A and B'.

Referee #3:

There are some concerns that remain unanswered:

1- How does the loss of non-canonical autophagy/ATG16L1 WD domain lead to the triggering of inflammatory cytokine production?

We have cited some recent work showing that the WD domain of ATG16L1 influences endocytosis of cytokine receptors, TLR4, CD36 and the β amyloid receptor. This can trigger inflammation.

P16 In 7. "A similar pro-inflammatory phenotype resulting from decreased trafficking of inflammatory cargoes is observed following disruption of non-canonical autophagy by LysMcre-mediated loss of Rubicon from macrophages or microglia^{6,20}. Studies also show that the WD domain of ATG16L1 can modulate endocytosis of cytokine receptors (Serramito-Gomez et al 2020). Interaction of the WD domain with the IL10 receptor (IL10-R), for example, promotes formation of endosomes containing IL-10/IL-10 receptor complexes leading to an enhanced anti-inflammatory signalling that would be lost in δ WD mice. Impaired recycling of Toll-like receptor 4, CD36 and the β -amyloid receptor TREM2 is also observed in microglia lacking the WD domain of ATG16L1 leading to neuroinflammation (Heckmann et al 2020)."

2- *What is the fate of LC3 during influenza A infection in mice lacking the WD domain of ATG16L1*

We do not know the answer to this question.

Thank you for submitting your revised manuscript. The study has been seen by two of the original referees, whose comments are shown below.

As you can see, while referee #2 finds that his/her criticisms have been sufficiently addressed, reviewer #1 has several remaining concerns that should be solved before I can officially accept your manuscript.

In addition, there are a few editorial issues concerning the text and the figures that I need you to address.

REFEREE REPORTS

Referee #2:

Overall, the authors have addressed most of my comments. I have some remaining remarks based on the newly added data:

The newly added LC3 blots in Figure 1C should have been performed in the presence/absence of Vps34 inhibition to distinguish canonical from non-canonical LC3 lipidation. Since this has been nicely shown in Figures 1A&B, the LC3 blots could be removed from Figure 1C while keeping the ATG16L1 blot (as well as including a loading control).

There are no statistical tests performed in Figure 1D and therefore the comparisons of LC3II levels between the samples and potential reduction in delta-WD40 cells during CQ/monensin treatment cannot be confirmed. The statement in pg.7 line 5 should be adjusted: "The LC3II signal in ΔWD MEFs incubated with monensin or chloroquine was however lower than controls". Similarly, the statement on pg.19: "western blot of LC3II (Fig 1C&D) did not reveal a statistically significant loss of canonical autophagy".

How was pixel intensity calculated in Figure 1H? This is not described in the legend or M&M.

The following statement on pg19 lines 4-6 also needs adjustment: " Nevertheless, it is not possible to exclude the possibility that removal of the WD and linker domains of ATG16L1 has a minor effect on conventional autophagy that might affect infection 'in vivo'." What is meant by "conventional autophagy" here? The authors, and previous work, show that the WD40 is dispensable for canonical LC3II lipidation. The WD40 may have functions in selective autophagy or in regulating ATG16L1 activity as indicated in subsequent sentences.

Figures 6A&C use different readouts to measure viral replication, where A uses PFU and B uses % infection. Can viral replication be directly compared using these two methods? If so, a justification of this should be included.

Figure 6D: is the change in viral dequenching significant in the quantification? The text (pg. 13) suggests a 2-fold increase in dequenching which is not apparent from this figure. Statistical quantifications are missing here. The FACS experiment in Figure 6E involves trypsinization of cells which could remove viral particles attached to the cell surface. Could this be the reason behind the apparent differences between panels D&E?

What is the conclusion from Figure 6? I am not sure the authors can exclude a potential effect on general endocytosis that is not specific to IAV viral fusion. A quick experiment using a general endocytic substrate (e.g. transferrin or dextran) could help support a more viral-specific defect in the delta-WD40 cells.

Referee #3:

The authors have addressed my concerns.

The authors have made all requested editorial changes.

ACCEPTED

8th Jan 2021

I am pleased to inform you that your manuscript has been accepted for publication in The EMBO Journal.

Corresponding Author Name: James P Stewart

Manuscript Number: EMBOJ-2020-105543R